# Bi-allelic variants in *RNF170* are associated with hereditary spastic paraplegia

Matias Wagner [1,2,3,20], Daniel P.S. Osborn [4,20], Ina Gehweiler[5,6], Maike Nagel[5,6], Ulrike Ulmer[5,6], Somayeh Bakhtiari[7,8], Rim Amouri[9,10], Reza Boostani[11], Faycal Hentati[9,10], Maryam M. Hockley[8], Benedikt Hölbling[5,6], Thomas Schwarzmayr[3], Ehsan Ghayoor Karimiani[4,12], Christoph Kernstock[13], Reza Maroofian[4], Wolfgang Müller-Felber[14], Ege Ozkan [4], Sergio Padilla-Lopez[7,8], Selina Reich[5,6], Jennifer Reichbauer[5,6], Hossein Darvish[15], Neda Shahmohammadibeni[15], Abbas Tafakhori[16], Katharina Vill[14], Stephan Zuchner[17,18], Michael C. Kruer[7,8], Juliane Winkelmann[1,3,19], Yalda Jamshidi [4,21] & Rebecca Schüle[5,6,21]*

Alterations of $Ca^{2+}$ homeostasis have been implicated in a wide range of neurodegenerative diseases. $Ca^{2+}$ efflux from the endoplasmic reticulum into the cytoplasm is controlled by binding of inositol 1,4,5-trisphosphate to its receptor. Activated inositol 1,4,5-trisphosphate receptors are then rapidly degraded by the endoplasmic reticulum-associated degradation pathway. Mutations in genes encoding the neuronal isoform of the inositol 1,4,5-trisphosphate receptor (*ITPR1*) and genes involved in inositol 1,4,5-trisphosphate receptor degradation (*ERLIN1, ERLIN2*) are known to cause hereditary spastic paraplegia (HSP) and cerebellar ataxia. We provide evidence that mutations in the ubiquitin E3 ligase gene *RNF170*, which targets inositol 1,4,5-trisphosphate receptors for degradation, are the likely cause of autosomal recessive HSP in four unrelated families and functionally evaluate the consequences of mutations in patient fibroblasts, mutant SH-SY5Y cells and by gene knockdown in zebrafish. Our findings highlight inositol 1,4,5-trisphosphate signaling as a candidate key pathway for hereditary spastic paraplegias and cerebellar ataxias and thus prioritize this pathway for therapeutic interventions.

[1] Institute of Human Genetics, Technische Universität München, Trogerstraße 32, 81675 Munich, Germany. [2] Institute of Human Genetics, Helmholtz Zentrum München, Ingolstädter Landstraße 1, 85764 Neuherberg, Germany. [3] Institut für Neurogenomik, Helmholtz Zentrum München, Ingolstädter Landstraße 1, 85764 Neuherberg, Germany. [4] Genetics Centre, Molecular and Clinical Sciences Institute, St George's University of London, London, UK. [5] Department of Neurodegenerative Diseases, Hertie-Institute for Clinical Brain Research and Center of Neurology, University of Tübingen, Hoppe-Seyler-Str. 3, 72076 Tübingen, Germany. [6] German Center for Neurodegenerative Diseases (DZNE), Otfried-Müller-Str. 27, 72076 Tübingen, Germany. [7] Barrow Neurological Institute, Phoenix Children's Hospital, Phoenix, AZ 85016, USA. [8] Departments of Child Health, Cellular & Molecular Medicine, Genetics, and Neurology, University of Arizona College of Medicine, Phoenix, AZ 85004, USA. [9] Neurology Department, Mongi Ben Hmida National Institute of Neurology, Tunis, Tunisia. [10] Neuroscience Department, Faculty of Medicine of Tunis, University Tunis El Manar, Tunis, Tunisia. [11] Department of Neurology, Mashhad, Iran. [12] Next Generation Genetic Clinic, Mashhad, Iran. [13] Centre for Ophthalmology, Institute for Ophthalmic Research, University of Tübingen, Tübingen, Germany. [14] Department of Pediatric Neurology and Developmental Medicine, Ludwig-Maximilians-University of Munich, Lindwurmstraße 4, 80337 Munich, Germany. [15] Cancer Research Center, Semnan University of Medical Sciences, Semnan, Iran. [16] Iranian Center of Neurological Research, Neuroscience Institute, Tehran University of Medical Sciences, Tehran, Iran. [17] Dr. John T. Macdonald Foundation, Department of Human Genetics, FL33136 Miami, USA. [18] John P. Hussman Institute for Human Genomics, University of Miami, Miller School of Medicine, FL33136 Miami, USA. [19] Munich Cluster for Systems Neurology (SyNergy), Munich, Germany. [20]These authors contributed equally: Matias Wagner, Daniel P.S. Osborn. [21]These authors jointly supervised this work: Yalda Jamshidi, Rebecca Schüle. *email: rebecca.schuele-freyer@uni-tuebingen.de

Disturbances in $Ca^{2+}$ signaling are emerging as a common pathophysiological pathway, and thus promising therapeutic target in a broad range of neurodegenerative diseases including Alzheimer's disease[1], Huntington's disease[2], and spinocerebellar ataxias (SCA)[3–5]. As a major intracellular $Ca^{2+}$ reservoir, the endoplasmic reticulum (ER) is essential for regulating intracellular $Ca^{2+}$ concentrations. Regulated $Ca^{2+}$ release from the ER is mediated by two types of $Ca^{2+}$ release channels: inositol 1,4,5-trisphosphate (IP3) receptors (IP3R) and ryanodine receptors (RyR). IP3Rs are large tetrameric complexes located in the ER membrane; they are activated by IP3 released from G-protein-coupled receptors in the plasma membrane. Activation results in efflux of $Ca^{2+}$ from the ER to the cytoplasm. Subsequently degradation of activated IP3Rs is mediated by the ER-associated degradation (ERAD) pathway[6]. Although the degradation of activated IP3R via the ERAD pathway is well understood, the basal turnover of IP3Rs is less clear with early studies suggesting lysosomal degradation of IP3Rs[7,8] as well as more recent support for involvement of the ubiquitin proteasome system[9].

A complex of the proteins erlin-1 and erlin-2, encoded by the genes *ERLIN1* and *ERLIN2*, are key components of the ERAD pathway, mediating ubiquitination of IP3Rs by the ubiquitin E3 ligase RNF170[10,11] and initiation of the proteasomal degradation of IP3Rs[12]. Mutations in *ERLIN1* and *ERLIN2* cause Hereditary Spastic Paraplegia (HSP)[13–20], a heterogeneous group of neurodegenerative motor neuron disorders (MND), primarily affecting the long motor axons of the corticospinal tract motor neurons and leading to the cardinal symptoms of progressive lower limb spasticity and weakness[21]. In complicated forms of HSP, neuronal systems other than the corticospinal tract are affected and spastic paraplegia is accordingly accompanied by additional neurological features such as seizures, cognitive deficits, ataxia, deafness, extra-pyramidal involvement, or peripheral neuropathy[21,22]. More than 100 genes are known to cause autosomal dominant, autosomal recessive, and X-linked forms of HSP; a subset of these genes have been cataloged by OMIM (www.omim.org) as Spastic Paraplegia Genes (SPG1–SPG80). Still, mutations in known HSP genes explain only about two-third of cases[21,23,24]. Mutations in novel HSP genes as well as novel mutation types that cannot be reliably detected or interpreted by current technology and prediction algorithms are likely to contribute to this 'missing heritability' in HSPs.

A specific founder mutation in *RNF170* has been associated with autosomal dominant afferent ataxia (ADSA) owing to degeneration of central sensory tracts, a phenotype unrelated to HSP, in two Eastern Canadian families[25–27].

Here, we show that mutations in *RNF170* are associated with autosomal recessive HSP in four unrelated families. Loss of RNF170 in patient-derived fibroblasts and knockout SH-SY5Y neuronal cell lines result in accumulation of the inositol 1,4,5-trisphosphate receptor that can be rescued upon RNF170 re-expression. In zebrafish, knockdown of *rnf170* leads to neurodevelopmental defects. Our findings highlight inositol 1,4,5-trisphosphate signaling as a candidate pathway for the development of future therapeutic interventions.

## Results

**Biallelic mutations in *RNF170* cause HSP.** In two siblings of an apparently autosomal recessive German family with early-onset HSP complicated by axonal peripheral neuropathy (family A, Fig. 1a) we performed whole genome sequencing (WGS) to identify the causative mutation, after extensive genetic testing for mutations in known HSP genes had failed to confirm the molecular diagnosis. We filtered for potentially biallelic rare coding and splice region variants and identified changes in five genes (*DNAH5, FCRL2, GPR98, RNF170, ZNF646*). Four of these could be excluded by segregation analysis in additional family members leaving only *RNF170*, encoding a ubiquitin E3 ligase (Supplementary Data 1).

The homozygous splice region variant in *RNF170* (NM_030954.3 [https://www.ncbi.nlm.nih.gov/nuccore/NM_030954.3]) c.396+3A >G is located within a haplotype shared between the apparently unrelated parents, pointing towards a potential founder effect consistent with the origin of both parents from the same small village in the Westerwald region in Germany. The c.396+3A>G change is predicted to result in loss of the splice donor site of exon 5 (Berkeley Drosophila Genome Project[28]). To confirm the splice effect we performed an RT-PCR of *RNF170* mRNA derived from peripheral blood and patient fibroblasts in patient A.4. RT-PCR revealed expression of a shortened transcript in both tissues, whereas the wildtype transcript could no longer be detected. Sequence analysis of the aberrant transcript demonstrated that this transcript lacked exon 5 (74-bp length), thereby leading to a shift of the reading frame (p.Ala109Asnfs*9). The aberrant transcript at least partially escapes nonsense mediated decay; expression of the aberrant *RNF170* transcript reaches 36/50% of normal *RNF170* mRNA expression levels in patient fibroblasts and peripheral blood, respectively (Fig. 1a–e). A truncated protein, however, which would be expected to be dysfunctional as it lacks the C-terminal half of the RING domain, could not be detected by western blot (Fig. 1f). Specificity of the antibody was confirmed by staining for RNF170 in a CRISPR/Cas9 knockout SH-SY5Y cell model.

**Identification of *RNF170* mutations in additional families.** In order to validate the association between biallelic loss-of-function mutations in *RNF170* and HSP we sought to identify further individuals carrying *RNF170* mutations using the web-based collaboration platform GeneMatcher[29]. In addition to the index case from family A, GeneMatcher returned three matches for *RNF170*, all categorized with an HSP phenotype (Table 1). In all families (families B–D), whole-exome sequencing (WES) had been performed and led to selection of *RNF170* as a potential candidate gene. Candidate variants and genes identified using a filter for (potentially) biallelic variants for each family are listed in the Supplementary Data 1. In family B, a consanguineous Baluch family from Iran, the homozygous missense variant c.304T>C, p.Cys102Arg segregated in the family including all four affected siblings with a LOD-score of 2.4 (Fig. 1g, h). The mutant residue lies in the RING domain of *RNF170*; the affected cystine is one of eight so-called zinc-organizing residues that collectively bind two atoms of zinc and thus maintain the rigid structure of the RING core domain[30]. In vitro mutation of Cys[102] has previously been shown to impair the ligase activity of RNF170 and suggested to act in a dominant-negative fashion[12].

In the Tunisian family C, trio WES was performed; analysis of copy number variations using ExomeDepth[31] and Pindel[32] detected a homozygous intragenic deletion of exons 4–7 of *RNF170* (Fig. 1i–m), resulting in the loss of not only the complete RING domain but also two out of three transmembrane domains. The variant was not seen in over 15,000 in-house controls as well as 60,000 exomes of the exome aggregation consortium database (as per August 2018). Breakpoint PCR and subsequent Sanger sequencing specified the InDel mutation as chr8:g .42,704,626_42,729,012delinsTTTTGGT (Fig. 1m). Screening of additional Tunisian index cases with pure and complicated forms of HSP ($n = 34$) for presence of this deletion revealed no additional cases (Supplementary Fig. 1).

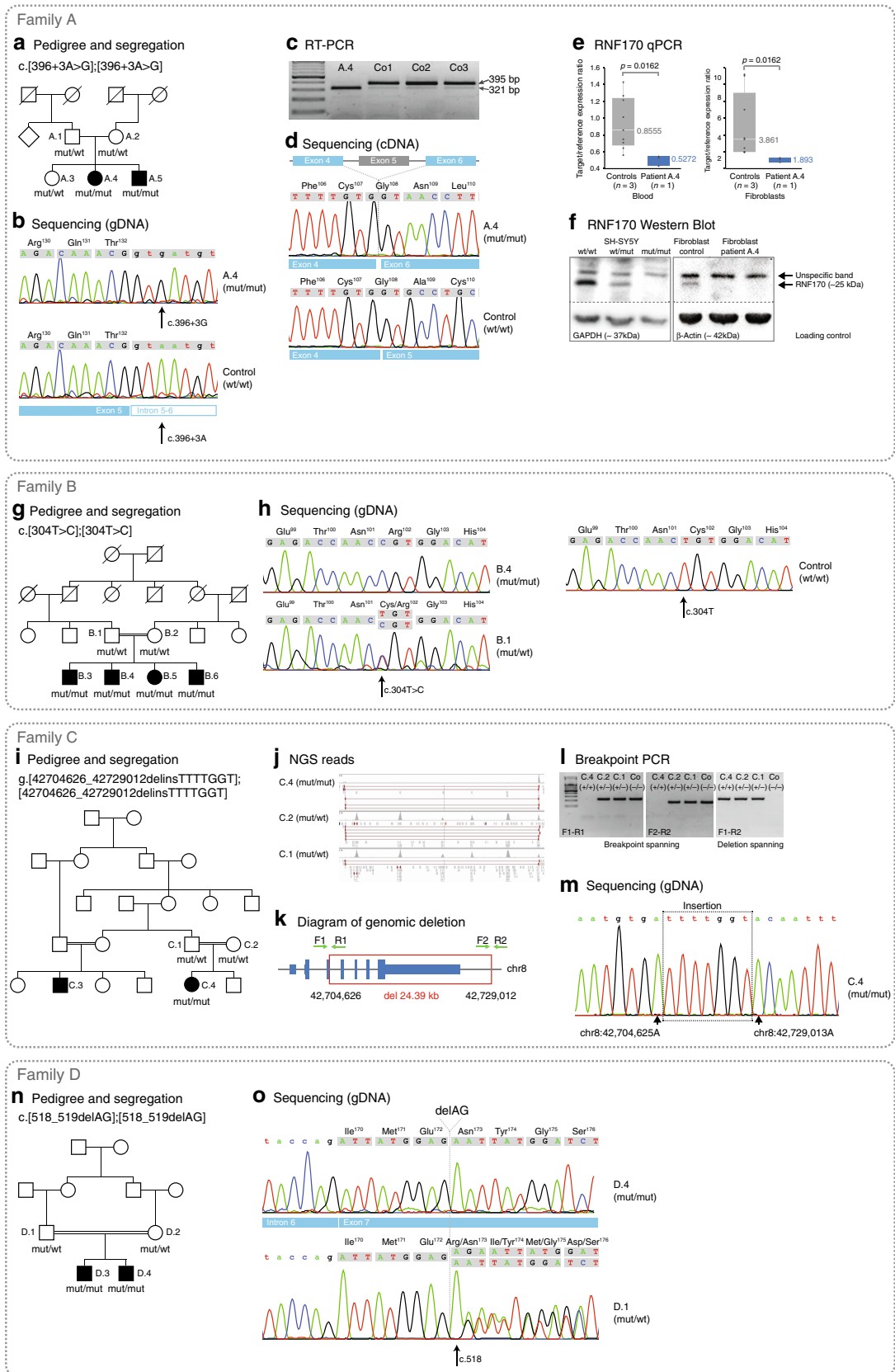

Finally, in two affected siblings of the consanguineous Iranian family D, a homozygous 2bp deletion was identified (c.518_519delAG), leading to a shift in the reading frame and introduction of a preterminal stop codon (p.Arg173Asnfs*49) (Fig. 1n–o). The predicted protein, if expressed, lacks both C-terminal transmembrane domains. All mutations showed complete co-segregation with the phenotype in the respective families (Fig. 1, Table 1).

**Fig. 1** Identification of biallelic RNF170 mutations in four families and functional characterization. **a–f** Identification of biallelic *RNF170* mutations in four families and functional characterization. **a** Pedigree of the family in which genome sequencing identified a homozygous splice region mutation in *RNF170* segregating with the disease. **b** Confirmation of the intronic variant c.396+3A>G in genomic DNA. **c** Gel electrophoresis and **d** consecutive Sanger sequencing confirmed the sole expression of a shorter transcript lacking exon 5 (wildtype transcript: 395bp; aberrant transcript: 321bp). **e** Quantitative real-time PCR from blood and fibroblast derived cDNA from individual A.4 demonstrated significantly reduced RNF170 expression in comparison with three control samples (Wilcoxon rank sum test, two-sided); **f** No residual RNF170 expression could be detected in patient fibroblasts. Note the unspecific band in the RNF170 western blot as well as the specific 25 kDa band corresponding to RNF170, that is abolished upon knockout of *RNF170* in SH-SY5Y cells. **g** Pedigree of family B and **h** variant confirmation by Sanger sequencing. **i** Pedigree of family C and segregation in the family. **j** The deletion was confirmed by visual analysis of split reads in the IGV browser. **k, l** In addition, primers were designed flanking the breakpoints as well as the deletion. **m** Subsequent Sanger sequencing of the breakpoint fragment was used to further characterize the variant. **n** The frameshift variant segregating in family D could be confirmed by **o** Sanger sequencing

**Clinical characterization of *RNF170*-related HSP**. Clinically, the most consistent finding among the nine affected individuals from four unrelated families that were available for a detailed clinical examination was lower limb predominant spastic paraparesis with mild upper limb involvement after longer disease durations (Table 1). Age of onset was invariably before the age of 5 with a median of 2 years. Optic atrophy was present in all seven cases that received a neuro-ophthalmological examination (Supplementary Fig. 2). Saccadic pursuit in families A and D as well as upper limb ataxia, ataxic gait, and cerebellar atrophy in B.3 and B.4 indicate that the cerebellum can be variably affected in *RNF170*-associated disease. Sensory evoked potentials revealed subclinical involvement of the central sensory tracts at least in later disease stages (A.4, A.5). Other features that were variably observed included mild cervical dystonia (A.4) and axonal sensorymotor peripheral neuropathy (A.4, A.5), findings consistent with the diagnosis of HSP.

**Mutations in *RNF170* result in accumulation of IP3R**. The nature of the mutations observed in families A, C, and D suggest a loss-of-function mechanism (Supplementary Table 1). In accordance with the hypothesis that loss of RNF170 results in reduced ubiquitination and proteasomal degradation of IP3R, basal levels of IP3R-3 were increased 2.2–3.8-fold in patient fibroblasts (Fig. 2a, b) compared with fibroblasts from healthy unrelated controls. In addition, degradation of IP3R upon stimulation of IP3 release with bradykinin in patient fibroblasts (A.4: c.396+3A>G, deletion of exon 5 / C.4: g.42704626_42729012delinsTTTTGGT) was completely abolished. In contrast, control fibroblasts demonstrated a stable decrease of IP3R subunit 3 (IP3R-3, main IP3R isoform in fibroblasts) levels to ~ 51% of baseline levels 60 min after bradykinin exposure (Fig. 2c, d).

Neurons, the primarily affected cell type in HSP, mainly express IP3R subunit 1 (IP3R-1). We therefore turned to a neuronal cell model to study the effect of deleterious *RNF170* mutations on IP3R-1 levels and degradation. Using CRISPR/Cas9, we introduced a homozygous 35-bp frameshift mutation into the neuroblastoma cell line SH-SY5Y; loss of RNF170 protein expression was confirmed by western blot (Fig. 1f). In concordance with our results obtained in patient fibroblasts, IP3R-1 accumulated in SH-SY5Y(RNF170$^{ko}$) cells with an increase of IP3R-1 levels to ~ 1.8 fold of SH-SY5Y(RNF170$^{wt}$) cells (Fig. 3a, b). This accumulation could be reversed by stable re-expression of wildtype RNF170 (SH-SY5Y(RNF170$^{ko}$(wt-HA)); "wildtype-rescue"), supporting causality of the RNF170 status for the observed IP3R-1 accumulation (Fig. 3a, b).

We then tested the effect of RNF170 deficiency on stimulus-dependent IP3R-1 degradation (Fig. 3c, d). Stimulation of wildtype SH-SY5Y cells with carbachol led to a mild decrease of IP3R-1 levels, which was most pronounced 2 h after stimulation (2 h: 79% of baseline); however, the response to stimulation was rather variable and changes over time were not

statistically significant (Dunnett's test for multiple comparisons with control ($t = 0$); p$^{2h} = 0.1722$, p$^{4h} = 0.7233$). In SH-SY5Y (RNF170$^{ko}$) cells, IP3R-1 degradation was completely abolished (2 h: 107% of baseline). The difference between SH-SY5Y (RNF170$^{wt}$) and SH-SY5Y(RNF170$^{ko}$) cells, however, did not reach statistical significance, (genotype*time: $p = 0.1956$; Fig. 3c, d; full-factorial repeated measures analysis). Even though these results have to be interpreted with caution, the data imply a trend towards normalization of IP3R-1 degradation by RNF170$^{wt}$ re-expression.

**Neurodevelopmental defects in *rnf170* knockdown zebrafish**. To further understand the function of *RNF170* during development, we turned to the zebrafish as a versatile model of vertebrate disease. The zebrafish orthologue, *rnf170* (NM_214750.1 [https://www.ncbi.nlm.nih.gov/nuccore/NM_214750]), shares 61% nucleotides and 63% of amino acids with the human *RNF170*-coding region or protein, respectively (Supplementary Fig. 3, 4). Sequence conservation between zebrafish and human suggests they may function similarly between species. To investigate how loss of *rnf170* activity affects development, we designed two non-overlapping morpholino oligonucleotides (MOs) against intron 2- exon 3 (E3MO) and intron 3- exon 4 (E4MO) of the zebrafish *rnf170* sequence in order to abrogate appropriate mRNA processing (Supplementary Fig. 5a). Microinjection of the morpholinos perturbed normal *rnf170* splicing, as identified through RT-PCR at 48 hpf (Supplementary Fig. 5b).

Knockdown of *rnf170* resulted in developmental defects visible by 48 hpf, these include microphthalmia, microcephaly, and loss of motility (Fig. 4a and Supplementary movies). These features are consistent with the expression of *rnf170* at 48 hpf, as observed through in situ hybridization. *rnf170* transcript was highly expressed in the brain and less so within intersomitic structures of the trunk (Supplementary Fig. 6a). Given the implications of RNF170 with neurodevelopment, we further evaluated the morphant phenotype through the analysis of acetylated tubulin staining, a neural marker. Neurogenesis in the cranium was remarkably reduced, specifically in the mid-hindbrain region (Fig. 4b). Transverse cranial sections at 4 dpf stained with haemotoxylin and eosin revealed structural differences in morphant brains compared with control embryos, with a distinct loss of ventricular cavities (Fig. 4d). Loss of movement, as determined by a touch-evoked motility assay (Supplementary movies 1–3) suggested motor neuron (MN) defects in morphant embryos. Indeed, immunoflourescent staining of MNs in the myotome revealed reduced antigen reactivity in 48 hpf morphant embryos compared to controls, acteylated tubulin staining appeared reduced and punctate, suggesting reduced MN function (Supplementary Fig. 6b). To evaluate whether MN defects were owing to delayed migration, embryos were further analyzed for acetylated tubulin in the myotome at 4 dpf. Morphants displayed persistent reduction in MN staining. The maintained expression

**Table 1 Clinical characteristics of RNF170 mutation carriers**

| ID | A.4 | A.5 | B.3 | B.4 | B.5 | B.6 | C.4 | D.3 | D.4 |
|---|---|---|---|---|---|---|---|---|---|
| Mutation | Ala109Asnfs*9 (hom) | Ala109Asnfs*9 (hom) | Cys102Arg (hom) | Cys102Arg (hom) | Cys102Arg (hom) | Cys102Arg (hom) | delEx4_7 (hom) | Arg173Asnfs*49 (hom) | Arg173Asnfs*49 (hom) |
| moi/gender | AR/F | AR/M | AR/M | AR/M | AR/F | AR/M | AR/F | AR/M | AR/M |
| Race/origin | Germany | Germany | Iran (Baluch) | Iran (Baluch) | Iran (Baluch) | Iran (Baluch) | Tunisia | Iran (Fars) | Iran (Fars) |
| Age at onset (y) | 3 | 5 | 2 | 2 | 2 | 2 | 2 | 3 | 3 |
| Age at exam (y) | 53 | 34 | 12 | 11 | 7 | 4 | 4 | 17 | 23 |
| Age at loss of independent walking | 20 | 22 | 11.5 | Still walking | Still walking | Still walking | Still walking | Sill walking | Still walking |
| Delayed motor development | - | - | + | + | + | + | + | - | - |
| Cognitive deficits | - | - | - | - | - | - | - | - | - |
| Visual system | Mild optic atrophy | Not examined | Severe optic atrophy | Moderate optic atrophy | Mild optic atrophy | Mild optic atrophy | Not examined | Optic atrophy | Optic atrophy |
| Oculomotor abnormalities | Saccadic pursuit | Saccadic pursuit | - | - | - | - | - | Saccadic pursuit | Saccadic pursuit |
| Dysarthria/dysphagia | -/- | -/- | +/+ | +/+ | +/- | +/- | -/- | +/- | +/- |
| UL/LL spasticity | +/+ | +/+ | -/+ | -/+ | -/+ | -/+ | -/+ | +/+ | +/+ |
| UL/LL tendon reflexes | Brisk/brisk | Brisk/brisk | Normal/brisk | Normal/brisk | Normal/brisk | Normal/brisk | Normal/brisk | Brisk/brisk | Brisk/brisk |
| UL/LL weakness | -/+ (proximal) | -/+ (proximal) | -/+ (distal) | -/+ (distal) | -/- | -/- | -/+ | -/+ | -/+ |
| Muscle atrophy | - | - | + (generalized, severe) | - | - | - | - | - | - |
| Extensor plantar response | + | + | + | + | + | + | - | + | + |
| Sensory deficits* | -/-/-/- | +/-/+/+ | -/-/-/- | -/-/-/- | -/-/+/- | -/-/+/- | -/-/-/- | -/-/-/- | -/-/-/- |
| Ataxia | - | - | + (upper limb and gait) | + (upper limb and gait) | + | + | - | + | + |
| Extrapyramidal involvement | Mild cervical dystonia | - | - | - | - | - | - | - | - |
| Urinary/fecal urgency or incontinence | +/- | +/- | -/- | -/- | -/- | -/- | -/- | -/- | -/- |
| Nerve conduction studies | Axonal polyneuropathy | Axonal polyneuropathy | Normal | Normal | n.d. | n.d. | Normal | Normal | Normal |
| Motor evoked potentials | n.d. | UL normal, LL reduced cortical amplitudes | n.d. | n.d. | n.d. | n.d. | n.d. | n.d | n.d |
| Sensory evoked potentials | LL no cortical potential (age 30) | UL prolonged central latency, LL no cortical potential (age 33) | n.d. | n.d. | n.d. | n.d. | Normal | n.d | n.d |
| Visually evoked potentials | n.d. | n.d. | Normal | Normal | Normal | n.d. | n.d. | Increased p100 latency and reduced amplitude | Increased p100 latency and reduced amplitude |
| MRI | n.d. | Cranium and cervical spine normal | Significant cerebellar atrophy | Cerebellar atrophy | Normal | n.d. | Cranium and cervical spine normal | Normal | Normal |

Moi mode of inheritance, UL upper limb, LL lower limb, y years, n.d. not done
* vibration/joint position/surface/temperature

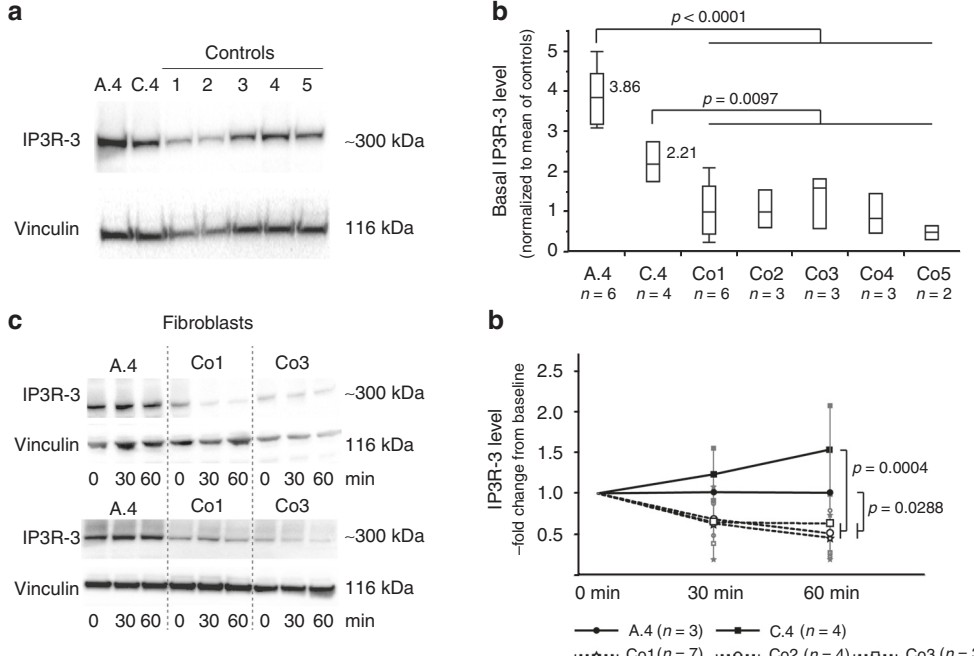

**Fig. 2** Loss of RNF170 results in decreased degradation of IP3R-3 in patient fibroblasts. **a** Immunoblot analysis of IP3R-3 in fibroblasts derived from individuals A.4 and C.4 shows increased expression levels in comparison with five controls (Co1, Co2, Co3, Co4, Co5). Western blots from a representative experiment are shown. **b** Semiquantitative immunoblot analysis indicates significantly increased (Tukey–Kramer HSD, t-sided) IP3R-3 expression. In the quantile blot, boxes indicate the 1st and 3rd quartile and median (center line); whiskers depict the 1st/3rd quartile ± 1.5* interquartile range). **c, d** IP3R-3 was activated by bradykinin stimulation of fibroblasts to trigger RNF170-dependent IP3R-3 degradation by the proteasomal system. IP3R-3 levels were assessed at baseline as well as 30 and 60 mins after stimulation. Physiological IP3R-3 reduction was observed in all three control cell lines (Co1, Co2, Co3), whereas levels were unaltered in patient-derived fibroblasts (derived from patients A.4 and C.4) (full-factorial repeated measures analysis; means and standard deviations are shown for each data point)

of acetylcholine receptors in morphant embryos suggests the muscle is primed for innervation, which fails with reduced Rnf170 function (Supplementary Fig. 6c). To validate specificity of the morpholinos we attempted to rescue with full length human RNF170. However, this resulted in exacerbation of the developmental phenotype. Disruption of endogenous expression by morpholinos, and global re-introduction of ectopic mRNA can sometimes result in severe phenotypes when the gene of interest is under tight spatial-temporal regulation[33,34]. Thus, the provision of a true rescue control here is likely to be impossible. For further validation, we therefore designed an additional morpholino against the translation start site (AUGMO). Congruently, injections of the AUGMO produced embryos with general morphology and motor neuron defects comparable with the splice morphants (Supplementary Fig. 7, Supplementary movies 4, 5). Taken together, these data support the role of Rnf170 in normal neurogenesis and importantly loss of *rnf170* in zebrafish recapitulates clinical features observed in the HSP patients.

To ascertain the functional relevance of variants identified in the patient cohort, human *RNF170* wildtype, the c.304T>C/p. Cys102Arg missense variant (family B), and p.Arg173Asnfs*49 truncated RNA (family A) was injected into wildtype embryos and phenotypes were assessed (Fig. 5). Embryo phenotypes were categorized as normal, mild, moderate, and severe. 50% of embryos injected with wildtype *RNF170* RNA showed a moderate to severe phenotype, which included truncation of the body axis and reduction in eye size, indicating toxic effects of *RNF170* wildtype overexpression. In contrast, no mock-injected control (MIC) embryos were categorized as either moderate or severe. Injections of variant containing RNAs showed results in line to what were observed in the mock-injected controls (Fig. 5a, b).

Eye size and embryonic length was then used as quantifiable features to be used for statistical analysis. One-way analysis of variance (ANOVA) using Tukey's multiple comparison test support the qualitative data: overexpression of wildtype *RNF170* significantly reduced embryonic length (wt RNA: mean 2227 μm ± SEM 202, $n = 16$ MIC: mean 2868 μm ± SEM 29.75, $n = 18$. Adjusted P value < 0.0001) and eye size (wt RNA: mean 34785 μm$^2$ ± SEM 2358. MIC: mean 47042 μm$^2$ ± 731. Adjusted P value <0.0001). No significant differences were observed between MIC and variant containing RNA injections. These data show that the variant RNAs are not as functionally active as wildtype RNF170 and support the identified genetic variants as disease causing (Fig. 5c, d).

## Discussion

We here report biallelic mutations in the ubiquitin E3 ligase gene *RNF170* as a likely cause of autosomal recessive HSP. The mutation types observed in the four families we describe in our study genetically support a loss-of-function mechanism. Further functional evidence that RNF170 deficiency may cause HSP via a loss-of-function mechanism is derived from functional studies that (i) demonstrated reduced expression of *RNF170* transcript and absence of RNF170 protein in patient fibroblasts (family A, Fig. 1f), (ii) increased basal levels and deficient stimulus-dependent degradation of IP3R-3 in patient fibroblasts-expressing mutant RNF170 protein (family A and C) as well as (iii) increased basal levels of IP3R-1 in neuronal SH-SY5Y cells and rescue by re-expression of RNF170$^{wt}$. Furthermore, *morpholino oligonucleotide* knockdown of *rnf170* in zebrafish led to neurodevelopmental defects and loss of motility, similar to other zebrafish models of HSP[35]. Although rescue experiments to

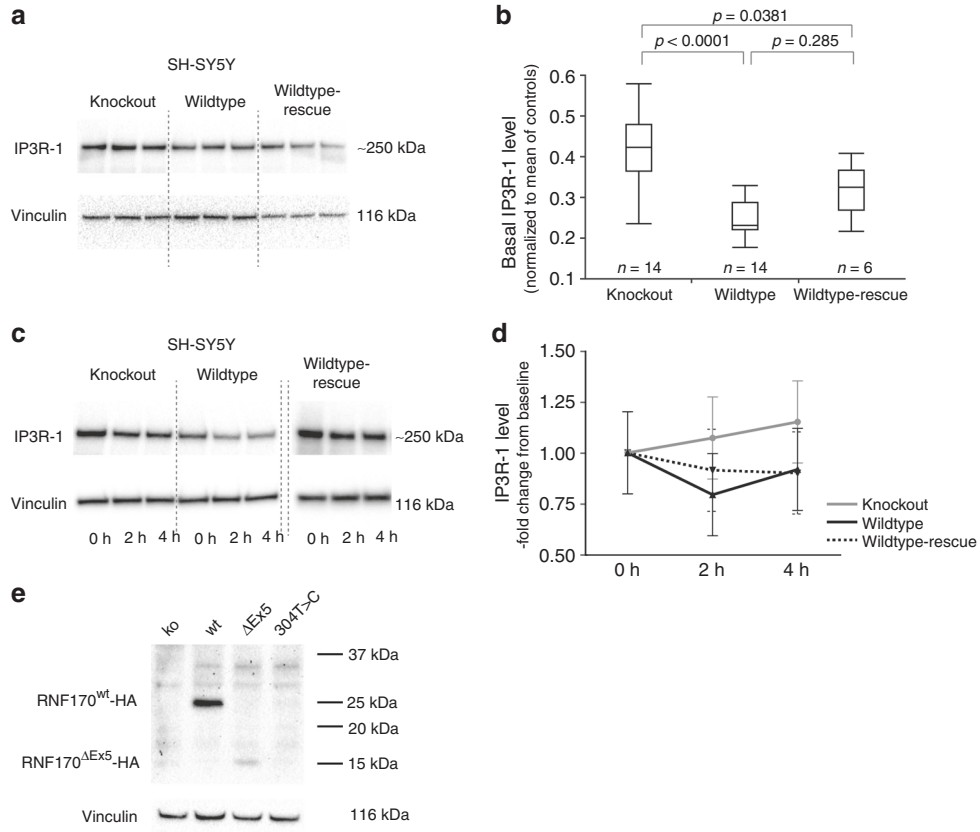

**Fig. 3** Effect of RNF170 mutations on IP3R-1 degradation and abundance in neuronal cells. **a, b** Immunoblot analysis of IP3R-1 in wildtype and knockout SH-SY5Y cells (SH-SY5Y(RNF170$^{wt}$)/$n = 14$ biologically independent samples; SH-SY5Y(RNF170$^{ko}$)/$n = 14$ biologically independent samples) and after re-expression of RNF170 in a knockout background (SH-SY5Y(RNF170$^{ko}$(wt-HA))/$n = 6$ biologically independent samples). SH-SY5Y(RNF170$^{ko}$) cells demonstrate significant accumulation of IP3R-1 (ko: mean 0.437 ± 0.133; wt: mean 0.247 ± 0.043) that can be rescued by re-expression of RNF170 (rescue: mean 0.318 ± 0.066; Tukey–Kramer HSD, two-sided). In the quantile blot, boxes indicate the 1st and 3rd quartile and median (center line); whiskers depict the 1st/3rd quartile ± 1.5* interquartile range. **c, d** IP3R-1 was activated by carbachol stimulation in neuronal SH-SY5Y cells, including wt and CRISPR/Cas9 generated RNF170 knockout cell lines (SH-SY5Y(RNF170$^{wt}$), SH-SY5Y(RNF170$^{ko}$) as well as SH-SY5Y cells stably expressing wildtype HA-tagged RNF170 in a knockout background (SH-SY5Y(RNF170$^{ko}$(wt-HA)). IP3R-1 levels were quantified by western blot at baseline ($t = 0$ h) and 2 h/4 h after stimulation. Neither the effect of the genotype on IP3R-1 degradation (wt vs. ko: $p = 0.1806$; repeated measures full-factorial analysis) nor the interaction between genotype and time was significant (wt vs. ko, genotype*time: $p = 0.1956$; repeated measures full-factorial analysis). Nine independent biological replicates were examined per genotype. Means and standard deviations are shown for each data point. **e** Expression of episomally expressed HA-tagged RNF170 was analyzed by immunoblot in SH-SY5Y cells

further prove specificity of the morpholino were unsuccessful, this is not unusual for endogenous genes subject to specific and complex spatial-temporal regulation[33,34].

The mechanism of IP3R-1 accumulation in neuronal SH-SY5Y cells is not entierly clear as we were not able to demonstrate a clear deficit of RNF170 deficient SH-SY5Y cells to degrade IP3R-1 upon stimulation with carbachol—in contrast to the strong defect in stimulus-dependent IP3R-3 degradation we observed in RNF170 mutant patient fibroblasts. Stimulation with carbachol triggered only a partial degradation of IP3R-1 in wildtype SH-SY5Y cells to ~80% of basal levels and the response was quite variable. It therefore remains to be determined whether the apparent IP3R-1 accumulation observed in SH-SY5Y(RNF170$^{ko}$) is the result of a defect in stimulus-dependent receptor degradation or disturbed basal turnover.

The missense mutation p.Cys102Arg observed in family B affects an amino-acid residue that, when mutated to serine in vitro in rat (corresponding rat amino acid: Cys101) leads to loss of ubiquitin ligase activity of RNF170, accumulation of IP3R and subsequent failure to degrade IP3R upon stimulation[12]. However, similar to other mutations affecting the RING domain of E3 ligases[36,37], the rat mutation p.Cys101Ser acts in a dominant-negative way at

least under conditions of overexpression in rat fibroblasts[12]. A dominant-negative mode of action, however, could not be confirmed for this variant in our zebrafish model. When overexpressing *RNF170*$^{Cys102Arg}$ in wildtype zebrafish, no adverse affects were noted on the morphology, whereas overexpression of wildtype *RNF170* led to morphological abnormalities including reduced embryonic length and reduced eye size. Although similar amounts of both wildtype and mutant RNA were injected, we cannot exclude the possibility that these differences are owing to reduced RNA stability rather than aberrant protein function of the mutant *RNF170*. The autosomal recessive mode of inheritance in family B with absence of features associated with HSP in heterozygous mutation carriers (e.g., B.1, B.2) as well as the absence of detectable expression of *RNF170*$^{Cys102Arg}$ after overexpression in SH-SY5Y (RNF170$^{ko}$) cells (Fig. 3e) also argue against a clinically relevant dominant-negative effect of the p.Cys102Arg mutation. A possible explanation for this discrepancy might be the extent of over-expression. Strong overexpression (via a CMV promotor in ref. [12]) might result in competition of mutant RNF170 for binding to the erlin-1/2 complex that may not be functionally relevant under in vivo conditions with equimolar amounts of wildtype and mutant RNF170.

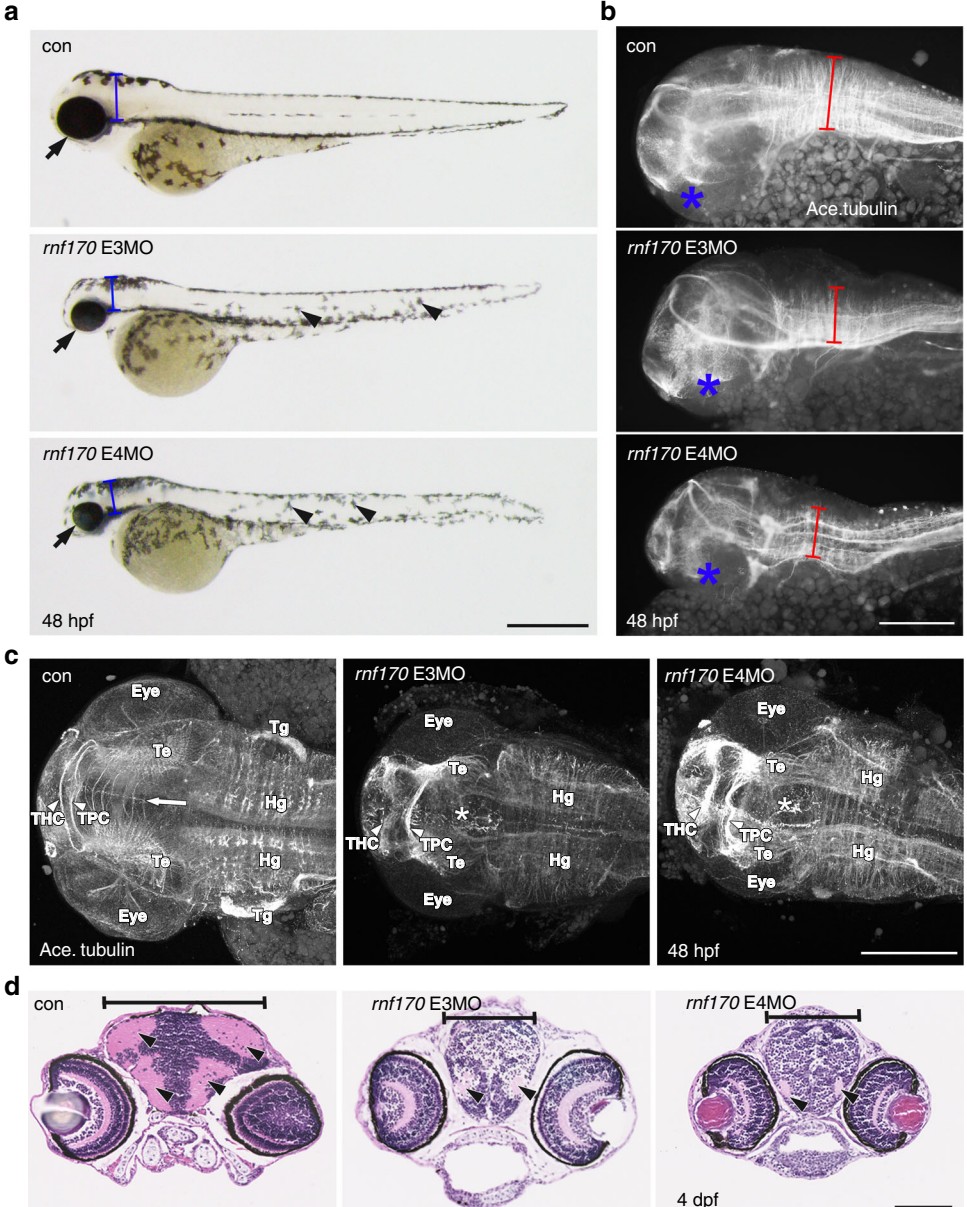

**Fig. 4** MO knockdown of *rnf170* results in morphological abnormalities, impaired neurogenesis, and motoneuron defects. **a** Representative images showing the morphology of live zebrafish embryos at 48 hpf injected with two different splice-blocking *rnf170* antisense MOs. *rnf170* morphants are characterized by a shortened body axis, micropthalmia (arrows), microcephaly (brackets), and alterations in pigmentation (arrowheads). Scale bar represents 500 μm. **b** Staining for the axonal marker acetylated tubulin at 48 hpf indicates impaired neurogenesis as shown by reduced neuronal density and migration (brackets) in the developing hindbrain of *rnf170* morphants, compared with control embryos. Asterisks indicate the position of the eye, scale bar represents 50 μm. **c** Dorsal flatmount images of acetyated tubulin stained embryos at 48 hpf showing loss of migrating axons across the intertectal commissure (arrow and asterisks), reduction of arborization in the tectum (Te), and thickening of the tracts of the habenular commissure (THC) and tracts of the posterior commissure (TPC) (arrowheads). Scale bar 200 μm. The eye, trigeminal glia (Tg) and hindbrain glia (Hg) are given as further landmarks. **d** Aberrant eye and brain development was observed in wax sections of *rnf170* morphants at 4 dpf stained with H&E. Reduction of cranial width (brackets) and ventricular cavities was apparent (arrowheads). Scale bar represents 100 μm

Phenotypically, we find that autosomal recessive HSP caused by RNF170 deficiency is characterized by infancy onset progressive spastic paraplegia, accompagnied by optic atrophy of variable severity and in some cases by cerebellar ataxia and subclinical involvement of the central sensory tracts. A missense mutation in *RNF170* (c.595C>T, p.Arg199Cys), going back to a common founder in the Eastern Canadian population, has previously been reported to cause autosomal dominant sensory ataxia (ADSA, MIM #608984 [https://www.omim.org/entry/608984]). ADSA manifests as late onset (4th–8th decade)

sensory ataxia owing to length-dependent affection of the central sensory tracts without clear involvement of the cerebellum or peripheral sensory nerves[25–27]. Although pyramidal signs were described in a subset of patients (pyramidal signs without manifest spasticity in 3/10 patients[27]), the overall ADSA phenotype bears little resemblance to the *RNF170*-associated HSP we describe here. Importantly, although the pathophysiology of ADSA is not completely understood, there are some fundamental differences on the molecular level between ADSA and *RNF170*-HSP pathophysiology. In both, *RNF170*-HSP as well as ADSA,

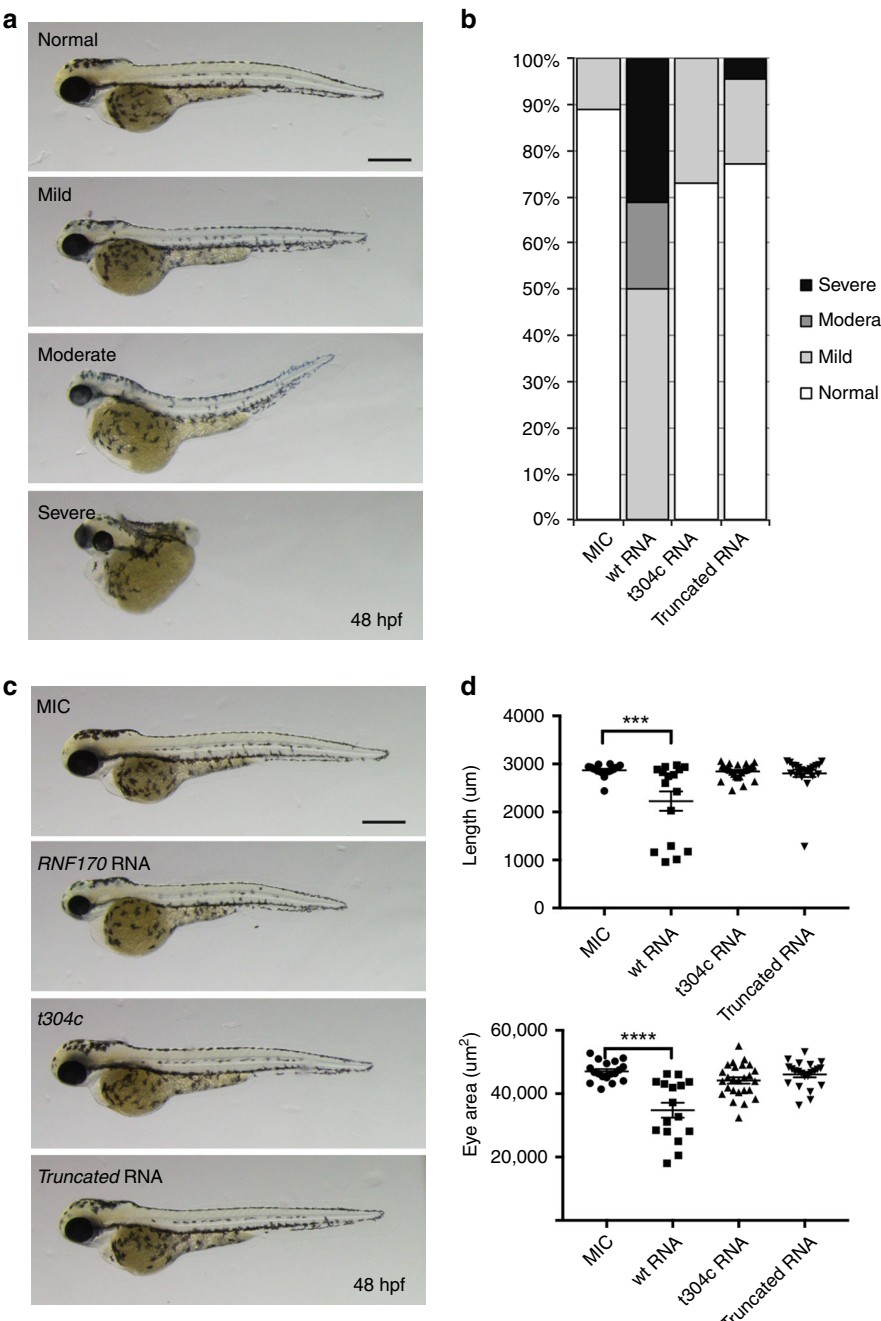

**Fig. 5** Overexpression of mutant *RNF170* in zebrafish. **a**, **b** Overexpression of wt but not mutant *RNF170* results in morphological abnormalities in zebrafish larvae. Representative images showing normal, mild, moderate, and severe morphology phenotypes in live zebrafish at 48 hpf. Overexpression of wt *RNF170* results in more severe phenotypes when compared with mock injected controls (MIC). Overexpression of truncated *RNF170* as well as mutant *RNF170* harboring the mutation c.304T>C, which was identified in family B, does not result in morphological abnormalities implying that the missense mutation results in a loss of protein function. Scale bar represents 400 μm. **c**, **d** Overexpression of wt but not mutant *RNF170* results in shortened body axis and smaller eye area. Representative images of MIC ($n = 18$) zebrafish larvae in comparison with overexpression of wt *RNF170* ($n = 16$) as well as mutants (c.304T>C, $n = 26$; and truncated RNA, $n = 22$) at 48 hpf. Only overexpression of wt *RNF170* but not mutant *RNF170* (both c.304T>C and truncated RNA) result in reduced body length and eye area in comparison with MIC embryos further delineating a loss of function effect of the mutation c.304T>C (one-way ANOVA with Tukey's multiple comparison test)

RNF170 protein levels have been shown to be decreased, albeit owing to distinct mechanisms. Although loss-of-function mutations lead to reduced *RNF170* expression in *RNF170*-HSP (here shown in patient fibroblasts (Fig. 1f) and SH-SY5Y cells (Fig. 3e)), RNF170[595C>T] levels in ADSA are decreased owing to increased auto-ubiquitination and proteosomal degradation of mutant RNF170. In *RNF170*-HSP, however, reduced RNF170 levels lead to an increase in basal IP3R levels and abolish IP3R degradation upon IP3 stimulation (Fig. 2) in patient fibroblasts. These findings are in accordance with previous studies in in vitro and in vivo RNF170 deficiency model systems, including demonstration of increased basal and stimulation-dependent IP3R levels in gonadotrophic αT3-1 pituitary cells upon RNAi depletion or CRISPR/Cas9 knockout of *RNF170*[9,12], and an increase of Itpr1 proteins

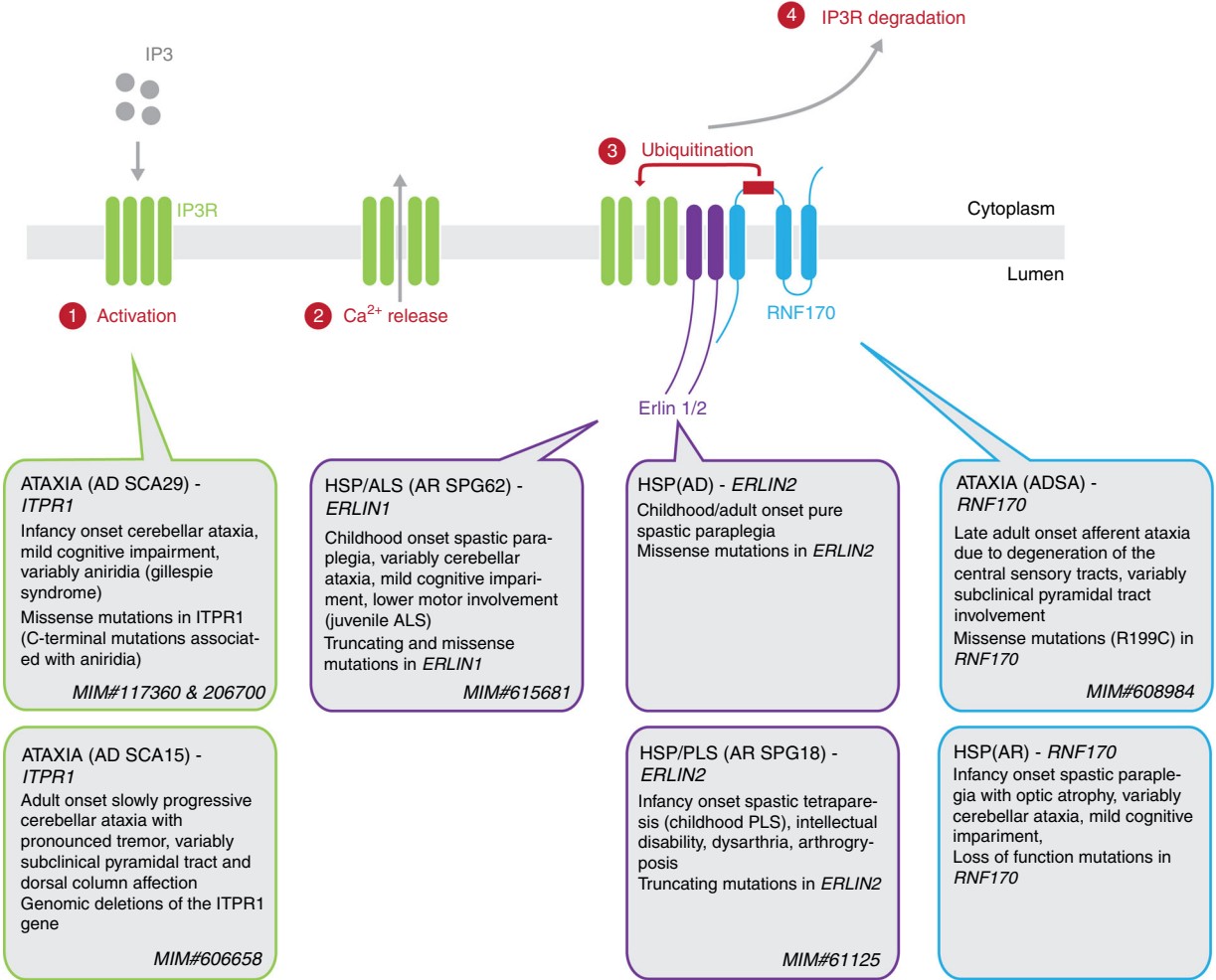

**Fig. 6** RNF170-dependent degradation of activated IP3R and genetic disorders affecting this pathway. Upon activation of the IP3R with IP3, calcium is released from the ER into the cytoplasm. This triggers the association of IP3R with the ERLIN1/2 complex leading to the ubiquitination of IP3R by the E3 ubiquitin ligase RNF170, resulting in the proteasomal degradation of IP3R. Mutations in all genes encoding components of this pathway are known to cause hereditary neurologic disorders, especially spastic paraplegia and spinocerebellar ataxia

(main neuronal isoform of the IP3R) in cerebellum and spinal cord of *Rnf170−/−* mice[38]. Most interestingly, *Rnf170−/−* mice develop age dependent gait abnormalities, which could resemble a HSP phenotype[38]. RNF170 deficiency might thus lead to increased IP3-dependent signaling via IP3Rs, followed by increased and potentially prolonged $Ca^{2+}$-release from the ER. In ADSA on the other hand, reduced levels of RNF170 do not translate into increased IP3R signaling, as IP3R levels are unaltered in patient lymphoblasts and $Ca^{2+}$ release from the ER is even decreased in this model contrary to expectations.[9] We suggest a toxic gain of function mechanism for the ADSA missense variant that is unrelated to transcript dosage effects; this hypothesis is supported by the dose-dependent toxicity of RNF170[Arg199Cys] in zebrafish larvae[26].

To put our findings into context, IP3R levels and thus IP3-dependent $Ca^{2+}$ release from the ER is tightly regulated by activity of the Erlin1/2–RNF170 protein complex. Genetic discoveries in recent years have emphasized the essential role of this pathway for function and maintenance of central motor neurons and Purkinje cells (Fig. 6). Mutations in *ITPR1*–genomic deletions (SCA15, MIM#606658 [https://www.omim.org/entry/606658]) as well as missense mutations (SCA29, MIM#117360)–have been shown to cause autosomal dominant cerebellar ataxia, that can be variably accompanied by aniridia (Gillespie syndrome, MIM#206700

[https://www.omim.org/entry/206700]). The latter can be caused by heterozygous variants acting in a dominant-negative fashion as well as biallelic loss-of-function mutations. Similar to *RNF170*, *ITPR1*-related disease is thus associated with both autosomal dominant and recessive inheritance. *ITPR1* deletions as well as at least some missense mutations lead to decreased $Ca^{2+}$ release from the ER upon stimulation in vitro[39,40], confirmed also in vivo in mice lacking two exons of the *ITPR1* gene (*ophisthotonos* mice)[41].

Truncating and missense mutations in *ERLIN1* have been associated with a range of phenotypes, from autosomal recessive childhood-onset HSP with variable cerebellar ataxia and mild cognitive impairment (SPG62, MIM#615681 [https://www.omim.org/entry/615681])[13] to amyotrophic lateral sclerosis[14]. Similarly, biallelic truncating mutations in *ERLIN2* cause infancy onset complicated HSP with lower limb predominant spastic tetraparesis, intellectual disability, pseudobulbar palsy and scoliosis (SPG18, MIM#611225 [https://www.omim.org/entry/611225])[15–17] as well as primary lateral sclerosis[18]. Two distinct missense mutations in *ERLIN2* (Thr65Ile, Ser129Thr) have been associated with autosomal-dominant pure HSP[19,20]. It has been shown recently that knockout of *ERLIN1* and *ERLIN2* both lead to an increase in basal IP3R-1 levels and impairment of IP3-dependent IP3R-1 degradation in gonadotrophic αT3-1 pituitary cells, changes that were also present in αT3-1 cells expressing *ERLIN2* carrying the pathogenic

missens mutant T65I[20] and similar to the alterations we observed in patient fibroblasts lacking *RNF170* (A.4, C.4) and neuronal RNF170 knockout cells (SH-SY5Y(RNF170[ko])).

Autosomal recessive and HSP-associated mutations in *ERLIN1, ERLIN2,* and *RNF170* as well as autosomal dominant missense mutations in *ERLIN2*—in contrast to the *RNF170* missense mutation reported to cause ADSA[9]—are thus all predicted to lead to an increase of basal IP3R levels and impairment of IP3R degradation. How this hypothesized increase in basal and stimulation-dependent IP3R levels would affect intracellular $Ca^{2+}$ handling and how phenotypic specificity of mutations targeting IP3 signaling is conveyed is currently unclear. Of note, however, genotype–phenotype correlation suggests that increased IP3 signaling is associated with an HSP phenotype while IP3 signaling seems to be reduced in ataxia[42]. The picture becomes even more complex when considering that dysregulated IP3-dependent $Ca^{2+}$ release from the ER has not only been implicated in *ITPR1*-related ataxias, but also a range of other neurodegenerative diseases including the autosomal dominant polyQ-expansion ataxias SCA2[3,4] and SCA3[5], Huntingtons disease[2,43], and Alzheimers disease[44,45]. IP3-dependent $Ca^{2+}$ signaling may thus be a prime target for therapeutic intervention in a wide range of neurodegenerative diseases.

## Methods

**Subjects.** The study was conducted in line with the Declaration of Helsinki and approved by the local institutional review boards at the University of Tübingen, Germany (054/2013BO1), the Technnical University Munich, Germany (5360/12), Next Generation Genetic Clinic (IR.MUMS.REC.1395.40), England, and Phoenix Children's Hospital, Phoenix, Arizona, USA (IRB # 15-080). All patients or their parents gave written informed consent for clinical data collection, collection and storage of biological samples, experimental analyses, and the publication of relevant findings. Patient consent covers sharing of biological samples under certain conditions; please contact the corresponding author.

**Exome and genome sequencing.** Exome and genome sequencing was carried out in DNA extracted from blood derived leukocytes. For exome sequencing, exonic regions were enriched using SureSelect Human All Exon XT V6 kits (Agilent, Santa Clara, USA) for family B and C and using xGen Exome Research Panel v1.0 (IDT, San Jose, USA) for family D. Genome-sequencing libraries for family A were prepared using TruSeq DNA PCR-Free Library Prep (Illumina, San Diego, USA). Paired-end sequencing was performed on HiSeq X HD v2.5 (family A), HiSeq2500 (family B), and HiSeq4000 (family C and D) platforms (all Illumina, San Diego, USA).

**NGS alignment and variant calling.** Reads were aligned to the UCSC hg19 (GCF_000001405.13 [https://www.ncbi.nlm.nih.gov/assembly/GCF_000001405.13/]) human reference genome using Burrows-Wheeler Aligner.[46] Single-nucleotide variants and small insertions and deletions were called using Freebayes (family A), GATK (family B and D) and SAMtools (family C)[47,48]. For a detailed description of the bioinformatical tools used see (Supplementary Table 2).

**Variant validation and breakpoint PCR.** For sequence validation and segregation analyses, the genomic loci of interest were PCR amplified and Sanger sequenced using standard protocols. PCR conditions are available upon request. For family C, breakpoint Sanger sequencing was used to confirm the variant identified by exome sequencing, determine the exact breakpoints of the deletion and for segregation analysis. In brief, a pair of primers (F1-R2, deletion spanning) was designed spanning the deletion and two pairs flanking the breakpoints (F1–R1 and F2–R2, breakpoint spanning) (Fig. 1k, l). The deletion spanning reaction results in a PCR product if the deletion is present at a heterozygous or homozygous state and the breakpoint spanning reactions yield PCR products when at least one wildtype allele is present. Oligonucleotide primer sequences are listed in Supplementary Table 3.

**Cell culture.** Primary fibroblast cell lines were grown from a 4–6 mm skin biopsy and were cultured in Dulbecco's Modified Eagle Medium (DMEM; Life Technologies, Carlsbad, USA) with 10% fetal bovine serum (FBS) and SH-SY5Y (ATCC CRL-2266) cells in DMEM/F12 (Life Technologies) supplemented with 15% FBS at 37 °C and 5% $CO_2$.

**RNA extraction, cDNA studies, and qRT-PCR.** RNA was isolated from whole blood collected into PAXgene Blood RNA System tubes (PreAnalytiX, Qiagen, Venlo, Netherlands) using PAXgene reagents according to the manufacturer's protocol. In fibroblasts, total RNA was prepared by using the High Pure RNA Isolation Kit (Roche Applied Science, Penzberg, Germany) according to manufacturer's instructions. RNA concentration and purity was determined using the NanoDrop ND1000 spectrophotometer (Thermo Fisher Scientific, Waltham, Massachusetts). Total RNA (500 ng) was reverse transcribed using Transcriptor High Fidelity cDNA Synthesis Kit (Roche Applied Science) according to manufacturer's instructions.

Gene expression was quantified by real-time PCR on the Real-Time PCR System on a LightCycler 480 device (Roche Applied Science). A melting curve was generated for each assay to check for specificity of the designed primers. Primer sequences are listed in Supplementary Table 3.

All PCR experiments were performed with three technical replicates. Gene expression of *RNF170* was quantified in relation to three reference genes, i.e., *RNF10, RNF111,* and *RPLP0*. For quantification, the advanced relative quantification module of the LightCycler software was used.

**Immunoblot analysis.** After cell lysis in RIPA buffer (Sigma-Aldrich, St. Louis, Missouri) including protease inhibitor (cOMPLETE Mini, Roche Applied Science), proteins were separated on a 3–8% NuPage Tris Acetate gel (IP3R-1 and IP3R-3, Thermo Fisher Scientific) or 12% Bis Tris gel (RNF170) and transferred onto a polyvinylidene difluoride membrane (IP3R-3 and RNF170; Immobilon, Merck Millipore, Burlington, Massachusetts) or nitrocellulose membrane (IP3R-1; Amerham Protrane Premium 0.45 NC, GE Healthcare, Chicago, USA). After blocking in non-fat dry milk TBS-T or Roche Block TBS-T, blots were probed with the primary antibody (rabbit anti-RNF170, Atlas Antibodies HPA054621 1:500; mouse anti-IP3R-3, BDBiosciences 610312, 1:1000; rabbit anti-IP3R, Abcam ab5804, 1:1000; mouse anti-β-Actin, Sigma A5441, 1:20000; mouse anti-Vinculin, Sigma V9131, 1:100000; mouse anti-GAPDH, Meridian H86504M, 1:10000), washed, incubated with the secondary antibody (Peroxidase AffiniPure Goat Anti-Mouse IgG (H+L) (115-035-003), 1:10000 and Peroxidase AffiniPure Goat Anti-Rabbit IgG (H+L) (111-035-003), 1:10000, Jackson ImmunoResearch, Cambridgeshire, UK), washed again and then developed with ECL solution (Immobilon Western HRP Substrat; WBKLS0500, Merck Millipore) on the ChemiDOC MP Imaging System (Bio-Rad).

**Stimulation-dependent IP3R degradation.** To stimulate IP3 release and thus IP3R, cells were treated with bradykinin (fibroblasts) or carbachol (SH-SY5Y). Prior to stimulation, cells underwent serum starvation. For this, cells were first washed in PBS (Sigma) and then cultured for 4 hours in DMEM (fibroblasts; Thermo Fisher Scientific) or DMEM/F12 (SH-SY5Y; Thermo Fisher Scientific) without FBS. Afterwards cells were treated with 300 nM bradykinin (fibroblasts; B3259, Sigma, powder dissolved in ddH₂O) or 1 mM carbachol (SH-SY5Y; C4382, Sigma, dissolved in ddH₂O) in the respective culture medium for $t = 0$, 30, 60 min (fibroblasts) or $t = 0$, 2, 4 h (SH-SY5Y). After treatment, cells were washed in PBS and scraped in adioimmunoprecipitation assay (RIPA) buffer (Sigma) including protease inhibitor (cOMPLETE Mini, Roche Applied Sciences) (fibroblasts) or PBS (SH-SY5Y). Immunoblots were then performed as described above.

**Generation of SH-SY5Y(RNF170[ko]) cells using CRISPR/Cas9.** To generate RNF170 knockout SH-SY5Y cells, the Synthego Gene Knockout Kit was used. To form RNP complexes, sgRNA and Cas9 protein were mixed in a ratio of 3:1. SH-SY5Y cells were cultured to 80% confluency. In total, $10^5$ cells were electroporated (AMAXA 2b, Lonza KitV, program G-004). Two RNP complexes containing two different sgRNAs (GAGGCUUGGGUGCAGGCAGAU and AGUGUAGAACUGC UGUCGAG) were simultaneous electroporated to obtain a 35-bp deletion causing a frameshift. After electroporation single cells were seeded on 10 cm dishes. Single-cell-derived colonies were picked manually and screened via PCR for presence of a deletion. Primer sequences are listed in Supplementary Table 3.

**Cloning of RNF170 constucts into neomycin selection plasmids.** To generate SH-SY5Y lines stably overexpressing *RNF170* mutants, the wildtype and mutant *RNF170* coding sequence (RNF170[wt]-HA, RNF170 [ΔEx5]-HA, RNF170[304T>C]-HA) was cloned into neomycin selection plasmids pSF-CMV-Ub-Neo/G418 AscI (Sigma-Aldrich), using the cloning sites *Bam*HI and *Hin*dIII.

**Generation of mutant SH-SY5Y lines.** To re-express wildtype and mutant *RNF170* in SH-SY5Y(RNF170[ko]) cells (see above), we electroporated $5 \times 10^6$ cells with 5µg plasmid (pSF-CMV-UB-NEO/G418 AscI-RNF170[wt]-HA /-RNF170[ΔEx5]-HA/-RNF170[304T>C]-HA) (AMAXA 2b, Lonza KitV, program G-004). One day after nucleofection medium was changed to selection medium, composed of DMEM/F12 + 15% FBS supplemented with 500 µg/ml G-418BC (A2912, Millipore). Henceforth, cells were cultured under these selection conditions. Presence of the plasmids was confirmed by Sanger sequencing (Supplementary Fig. 8, Supplementary Table 3).

**Zebrafish experiments.** All zebrafish studies were conducted in compliance with all relevant ethical regulations for animal testing. The studies were approved by the local (St George's University of London) institutional review board. Wildtype (AB × Tup LF) zebrafish were used for all zebrafish experiments. Antisense MO oligonucleotides (Genetools, LLC) were designed against the translational start site

(AUGMO: CCATCACTGCTGATCATGTCATG), Intron 2-Exon 3 (E3MO: CGCTCCTGATGGAGGAAAACACACG) and Intron 3-Exon 4 (E4MO: CACCTGATGGAGAGACACAGCGTTA) splice sites of zebrafish *rnf170*. Morpholinos were injected into embryos at the 1–2 cell stage and incubated at 28.5 °C untiled the desired stage. A control morpholino was used for comparison, targeting an intronic sequence in the human beta-globin gene. Specificity of the splice morpholinos was confirmed by RT-PCR. RNA was extracted from 30 embryos per experimental group at 48 hpf using TRIzol (Invitrogen, Thermo Fisher Scientific) as described in ref. [49]. First-strand cDNA was synthesized using random nanomers (Sigma-Aldrich) and omniscript transcriptase (Qiagen), according to manufacturer's instructions. Standard PCR was performed using primers surrounding the Intron 2-Exon 3 splice site (E2F: GATCAGCAGTGATGGAGGGG, I2R: CGTGTGTGTAAGAGAGAGAGTGT, E3R: CTCCTGACTCTCTGGGTGGA) and Intron 3-Exon 4 splice site (E3F: TCCACCCAGAGAGTCAGGAG, I3R: CTGATGGAGAGACACAGCGT, E4R: GTGTCCGCAGTTGGTCTCAA). For *RNF170* rescue experiments, site-directed mutagenesis was performed using Agilent's QuickChange II kit on *RNF170* cloned into BamH1 and Not1 sites of the pCS2+vector. PCR amplification to add SP6 promoter and short 3' polyadenylation site was performed using the following primers: SP6 forward ATTTAGGTG ACACTATAGAATGTACCCATACGATGTT and sPA reverse CATTTCGTAT TTTATTTTCATCTAGTTAGCCTTTGG. RNA was transcribed using the SP6 ambion MAXIscript kit, following the manufacturer's instructions. Approximately 100 pg of RNA was injected into wildtype embryos.

In situ hybridization was performed using standard protocols by cloning the full-length zebrafish *rnf170* into pGEMTeasy (Promega). Larvae were fixed with 4% paraformaldehyde, embedded in wax, and sectioned followed by staining with hematoxilin and eosin.

Wholemount immunohistochemistry was conducted using primary antbodies against acteylated Tubulin (Sigma-Aldrich, T6793) and alpha Bungarotoxin (Thermo Fisher, B13422) at 1:500 and 1:100 concentrations, respectively, combined with appropriate secondary antibodies (Invitrogen, Thermo Fisher Scientific) used at 1:1000.

**Statistical analysis**. To compare continuous variables (e.g., IPR3-R levels in patient fibroblasts/SH-SY5Y cells, body length, and eye area in zebrafish embryos) across groups, a one-way ANOVA, followed by Tukey–Kramer HSD post hoc testing was used. To compare the response to bradykinin/carbachol stimulation in fibroblasts or SH-SY5Y across genotypes, we performed a full-factorial repeated measures ANOVA with subject ID as a random effect and mutation status and time as fixed effects. Statistical analysis was performed using Jmp14.2 for Mac.

**Reporting summary**. Further information on research design is available in the Nature Research Reporting Summary linked to this article.

## Data availability

The authors declare that all data supporting the findings of this study are available within the paper and its supplementary information files. Whole-genome data sets for family A are available to all registered users to the RD-Connect platform (https://platform.rd-connect.eu) via publication of the Solve-RD data collection (http://solve-rd.eu); for the remaining families consent restrictions preclude sharing of full data sets; only specific information (e.g., secondary variants etc. but not full data sets) can be obtained upon request from the corresponding author. The source data underlying the Fig. 1c, e, f, Fig. 2, and Fig. 3 are provided as a Source Data file.

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

## Acknowledgements

We thank the patients and their families for participation in this study. We thank the Helmholtz-Zentum Munich NGS core facility, especially Elisabeth Graf and Tim Strom as well as Gertrud Eckstein, Peter Lichtner, and Veronika Schwarzbauer for their excellent support. We are also grateful to Lisa Abreu and Matt Danzi from the Hussman Institute for Human Genomics in Miami for their expert support with handling of WGS samples and data files and we thank Katrin Dillmann from the University of Tübingen for excellent clinical coordination of this study. This study was supported by the E-RARE JTC grant "NEUROLIPID" (BMBF, 01GM1408B to R.S.), the Horizon 2020 research and innovation programm via grant 779257 "Solve-RD" to R.S. and via the ERA-NET Cofund action No. 643578 by the BMBF under the frame of the E-Rare-3 network "PREPARE" (01GM1607: S.R. and associated partners S.Z., F.H.) and the STC-TUNGER-2015 grant "TUNGER-GENE" (01DH16024: R.S., R.A., F.H.) and via funding for the translational research consortium for HSP TreatHSP (01GM1905 to R.S.), the National Institute of Health (NIH) (grant 5R01NS072248 to R.S. and S.Z., grants 1R01NS075764, 5R01NS054132, 2U54NS065712 to S.Z., grants NS083739, 1K08NS083739 and 1R01NS106298 to M.C.K.), the Doris Duke Charitable Foundation (grant CSDA2014112 to MCK), a Valley Research Partnership award (SB) and the Interdisciplinary Center for Clinical Research (IZKF) of the University of Tübingen Medical School (scholarship 2017-1-16 to I.G.). We acknowledge support by Deutsche Forschungsgemeinschaft and Open Access Publishing Fund of University of Tübingen. R.M. would acknowledge the Queen Square Genomics group at University College London, which is supported by the National Institute for Health Research University Collegel London Hospitals Biomedical Research Centre.

## Author contributions

Design and conceptualization of the study: R.S., D.P.S.O., M.W., Y.J. Acquisition of clinical data: R.A., R.B., S.B., H.D., W.M.-F., F.H., C.K., E.G.-K., M.K., N.S., R.S., A.T., K.V., F.H. Acquisition of experimental data: DPSO (zebrafish studies); R.A., S.B., I.G., M.M.H., Y.J., M.K., R.M., E.O., S.P.-L., R.S., J.W., M.W., T.S., S.Z. (genetic studies); I.G., B.H., M.N., U.U., S.R., J.R., R.S. (cell lines and patient tissues). Analysis and interpretation of experimental data: S.B., H.D., I.G., M.M.H., Y.J., R.M., D.P.S.O., E.O., B.H., S.P.-L., M.N., S.R., J.R., R.S., A.T., U.U., J.W., M.W. Drafting of the manuscript: R.S., D.P.S.O., M.W., Y.J. Revising the manuscript for important intellectual content: A.L.L. Final approval of the version to be published: A.L.L. Agreement to be accountable for all aspects of the work: R.S.

## Competing interests

The authors declare no competing interests.
