## [Peer Review File · Nature Communications]

Reviewers' Comments:

Reviewer #1:

Remarks to the Author:

Wagner, Osborn and colleagues report a new causative gene for a recessive HSP/neuropathy/ataxia-like condition. Four different mutant loss-of-function alleles in the RNF170 gene are described in four separate families – the multiple mutations provide considerable support for the gene identification, in addition to the extensive bioinformatic analysis conducted on each individual mutation. Morpholino knockdown and overexpression studies in zebrafish support a role for the identified gene in neuronal development/differentiation (although without evidence for roles in degeneration), and support the model that the identified mutations affect RNF170 gene function.

A further potential strength of the paper is that it suggests plausible patho-mechanisms for RNF170, via impaired degradation of IP3 receptors, which would be predicted to potentiate Ca²⁺ signalling in mutant cells and in patients. This mechanism would link RNF170 function to that of many other degenerative conditions and causative mutations, thus opening routes to therapies for the conditions described here, as well as providing additional targets for other conditions in which ER and Ca²⁺ function is implicated. Therefore, evidence for this model would make the paper a significant mechanistic advance – although this evidence is not yet well developed.

Strengths:

- 4 affected families, 4 different mutant alleles in RNF170, with strong molecular analyses of the mutations
- Strong predictions that mutations are severe loss-of-function alleles (in one case affecting splicing and protein expression) and potentially causative
- Some evidence that one individual has expected cellular phenotype of lower IP3R degradation (but only n=3 repeats, one individual)
- MO treatment of zebrafish causes neuronal developmental defects (although not precisely defined)
- Over expression of WT gene, but not (to the same extent) mutants, causes developmental defects.

Weaknesses:

- While the work is good as far as it goes, the major weakness is the very limited evidence that pathogenic mutations block degradation of IP3 receptors in patients– only three datapoints from cells taken from one patient. I appreciate the constraints of obtaining cells from additional patients, but if this is not possible, the proposed mechanism would be greatly strengthened by additional evidence – for example by generating homozygous mutant human cells in a different genetic background, zebrafish, Drosophila, or another model carrying the same amino acid change as in Fig 2B.
- Legends and figure layout sometimes poor – particularly the complex Fig 1.

For example:

- Links between legend and panels are not intuitive. For a start it would be easier to find the data referred to in each legend, if panels within each part of the figure, A, B, C, D, were explicitly subdivided as i, ii, iii, etc. Also many features in the legend are not explicitly labelled; many features in the figure are not explained in the legend, e.g. sizes of bands.
- Also, different parts of the figure are not clearly separated, e.g. B iii (in the above numbering) might easily belong to A iii.
- Fig. 1 and elsewhere: Use of +/- for homozygous mutant individuals is counterintuitive and confusing for many readers: -/- is conventional in every other organism. If the authors wish to use this counterintuitive convention, they should explain it explicitly in their Fig legend.
- When coding regions are shown with sequencing chromatograms, please add DNA and amino

acid coordinates to help the reader relate their data to their interpretation in the text.
Therefore: Please review the layout and legends of all figures and supplementary figures to improve the annotation and legends.

Fig 2d: Please add some indicator of variability to graph, e.g. SEM and N (and ideally individual datapoints if not too many)

Fig 3 and Supp Fig 5: MO knockdown efficiency should be quantified. Also please state in Fig 3 legend what the control MO was.

Fig 4 graphs. Please show N. And please replace bar graphs by graphs that show individual datapoints as in Fig 2b; mean and SEM (if normally distributed) can be superimposed on these as horizontal lines.

Abstract:

Line 68: those -> some

Line 73

Introduction:

Line 84: Regulated Ca²⁺ release from the ER is mediated...

Line 87: G-protein-coupled is hyphenated

Reviewer #2:

Remarks to the Author:

This paper reports that loss of function mutations in RNF170 are linked with Hereditary Spastic Paraplegia (HSP) in several families. RNF170 is a ubiquitin E3 ligase, and InsP3R3 is one of its known targets. This genetic finding in itself is interesting and worth reporting, and genetic portion of the paper appears to be well done and convincing.

To explain the linkage between RNF170 and HSP, the authors argue that loss of function of RNF170 resulted in enhanced expression of InsP3Rs and abnormal calcium signaling in cerebellar Purkinje cells and in other types of neurons as well. To support these claims, they include data with fibroblasts obtained from one of the families and demonstrate that stimulus-induced degradation of InsP3R3 in these cells is impaired and steady-state levels of InsP3R3 are elevated several fold (Fig 2). These are interesting data, but it is InsP3R1 and not InsP3R3 is a predominant neuronal isoform highly enriched in Purkinje cells. InsP3R1 in Purkinje cells does not undergo stimulus-dependent degradation, and in my opinion extrapolating the data with InsP3R3 in fibroblasts to InsP3R1 in Purkinje cells is too speculative. The authors have to establish how loss of RNF170 affects expression levels and function of cerebellar InsP3R1. Most likely such experiments will require creation and analysis of RNF170 knockout mice.

They also present some data with Zebrafish, in which splicing of RNF170 was abrogated by morpholinos. Some interesting results were obtained, but these appear to be mostly developmental defects, the relevance of these findings for HSP is not clear. It is also not known if these defects are due to enhanced function of InsP3R or due to some other targets of RNF170. Genetic interaction experiments were not performed. It would have been informative to find out if InsP3R knockdown rescues RNF170 knockdown phenotype in Zebrafish. It was not done.

They also perform some mutant RNF170 overexpression experiments in zebrafish embryos, but these are very difficult to interpret. If they wanted to test different RNF170 mutants perhaps better approach will be to perform rescue experiments with RNF170 KO SHY cells.

Overall, my opinion is that this paper potentially very interesting but not sufficiently developed.

Genetic linkage with RNF170 LOF mutations is convincing. Proposed mechanism that involved upregulation of neuronal InsP3R1 function needs to be established further, most likely using RNF170 KO mouse model.

Reviewer #3:

Remarks to the Author:

This paper claims that inositol 1,4,5-trisphosphate signalling should be regarded as a key pathway for therapeutic intervention in hereditary spastic paraplegias and ataxias. Since ITPR1, ERLIN1 and ERLIN2 are already associated with these phenotypes, RNF170 expands the relevance of this important pathway to this disease category.

The conclusions of the paper are original in that they describe a new mode of inheritance and a slightly different phenotype than the one already associated with mutations in RNF170. This paper is of importance in the field since (1) it provides an independent confirmation of the involvement of RNF170 in neurodegenerative diseases (2) it associates mutations in RNF170 with an other mode of inheritance (3) it describes RNF170 mutations in a wider geographic and ethnic context (4) It expands on the biological validation of RNF170 and the understanding of its role in neurodegenerative diseases.

Overall, RNF170 had been associated so far with a limited phenotype (autosomal dominant sensory ataxia, a disease of the cerebellum and the posterior columns). Following the publication of the present paper, it will become necessary to include RNF170 in HSP and ataxia panels, a key finding to maintain comprehensive clinical genetic testing.

The overall quality of the scientific work involved in this paper is very high, considering the standards required for the validation of genetic findings (WES, Sanger confirmation, segregation of variants, replication in other populations, fibroblasts, zebrafish model).

One critique is a matter of context and appraisal. References 31-32-33 are not mentioned in the introduction, and ADSA is not mentioned in the abstract. This omission does not allow the reader to place the findings in their context. RNF170 has been clearly shown to cause ataxia in the context of a spinal cord disease, and RNF170 is already known therefore as an important pathway for development of therapeutic interventions in this disease category.

Lastly, Figure 5 would be better suited in a review paper.

Nicolas Dupré, MD, MSc, FRCP(C)
Neurologist & Clinician Scientist
CHU de Québec - UL
Associate Professor
Faculty of Medicine, Université Laval

Reviewer #4:

Remarks to the Author:

The manuscript "IP3 Receptor Degradation – A Mutational Hotspot for Hereditary Ataxia and Motor Neuron Disease" by M. Wagner is a well conducted genetic study with a initial functional characterization of the consequences of the identified mutations in cells and zebrafish.

The study is well presented and discussed and provides strong genetic evidence for the involvement of IP3 signaling in the pathophysiology of HSP. However there are oe major and

several some minor things to consider and some controls to include to validate and support the conclusions:

First and most importantly, the lack of a phenotype upon mRNA of the mutations t304c and truncated RNA could just simply reflect unstable RNA expression in zebrafish. Equal amount of protein translation needs to be shown here, eg by injection of tagged mRNAs and subsequent Western blot against the tag with quantification of the amount of protein translated.

Second, it should be pointed out that only about 50 of the transcripts are affected by the morpholinos.

Third, in figure 4 d the motor neurons seem to be oriented from posterior to anterior or the picture is aligned in a different way than the top 2 panels. The finding of more punctate staining with the ac tub antibody in the morpholino injected embryos could also reflect differences during fixation and should not be overstated.

Fourth, phenocopy by overexpression of IPR3R-3 in zebrafish would be a nice addition.

Reviewers' comments:

Reviewer #1 (Remarks to the Author):

Wagner, Osborn and colleagues report a new causative gene for a recessive HSP/neuropathy/ataxia-like condition. Four different mutant loss-of-function alleles in the RNF170 gene are described in four separate families – the multiple mutations provide considerable support for the gene identification, in addition to the extensive bioinformatic analysis conducted on each individual mutation. Morpholino knockdown and overexpression studies in zebrafish support a role for the identified gene in neuronal development/differentiation (although without evidence for roles in degeneration), and support the model that that the identified mutations affect RNF170 gene function.

Reply: To extend our study further, we evaluated the zebrafish motoneurons (MNs) at a later time point (4 dpf). In support of a degenerative role for RNF170, we find that morphant embryos show progressive loss of MNs at this later stage. (See supplementary Fig 7c).

A further potential strength of the paper is that it suggests plausible patho-mechanisms for RNF170, via impaired degradation of IP3 receptors, which would be predicted to potentiate Ca²⁺ signalling in mutant cells and in patients. This mechanism would link RNF170 function to that of many other degenerative conditions and causative mutations, thus opening routes to therapies for the conditions described here, as well as providing additional targets for other conditions in which ER and Ca²⁺ function is implicated. Therefore, evidence for this model would make the paper a significant mechanistic advance – although this evidence is not yet well developed.

Strengths:

- 4 affected families, 4 different mutant alleles in RNF170, with strong molecular analyses of the mutations
- Strong predictions that mutations are severe loss-of-function alleles (in one case affecting splicing and protein expression) and potentially causative
- Some evidence that one individual has expected cellular phenotype of lower IP3R degradation (but only n=3 repeats, one individual)

Reply: To provide additional evidence that the cellular phenotype of impaired IP3R degradation is indeed a consistent finding in RNF170-HSP we have now analyzed IP3R degradation in two unrelated patient fibroblasts lines (from families A and C) as well as a neuronal model system which we generated by knocking out RNF170 in SH-SY5Y cells using CRISPR/Cas9 (updated Fig. 2, new Fig. 3). We were able to confirm the IP3R degradation deficiency in all tested lines (see details below) and are therefore confident that impaired IP3R degradation is a consistent finding in RNF170-HSP. For further details see comments below.

- MO treatment of zebrafish causes neuronal developmental defects (although not precisely defined)

Reply: We now present additional dorsal flatmount images of morphants stained with acetylated tubulin which show a more complete detailed annotation of the head defects (Fig. 4C).

- Over expression of WT gene, but not (to the same extent) mutants, causes developmental defects.

Weaknesses:

- While the work is good as far as it goes, the major weakness is the very limited evidence that pathogenic mutations block degradation of IP3 receptors in patients— only three datapoints from cells taken from one patient. I appreciate the constraints of obtaining cells from additional patients, but if this is not possible, the proposed mechanism would be greatly strengthened by additional evidence – for example by generating homozygous mutant human cells in a different genetic background, zebrafish, Drosophila, or another model carrying the same amino acid change as in Fig 2B.

Reply: As stated above, we have now added analyses in a second primary fibroblasts line (family C) carrying a different RNF170 mutation and also confirmed our findings in neuronal SH-SY5Y cells with CRISPR/Cas9 generated knockout of the RNF170 gene. In the latter lines we also stably overexpressed three different RNF170 mutations (mutations observed in families A and C as well as the previously published mutation leading to autosomal dominant sensory ataxia (Valdmanis et al. 2011)). Using these cell lines, we were able to confirm that

- RNF170 mutations observed in HSP patients (A.4 (c.396+3A>G, deletion of exon 5) and C.4 (c.304C>T, p.Cys102Arg)) lead to increased basal levels and impaired IP3-dependent degradation of IP3R-3 in patient fibroblasts
- Genetic knockout of RNF170 (CRISPR/Cas9) leads to impaired degradation of IP3R-1 in the neuronal cell line SH-SY5Y compared to wildtype cells. This defect can be rescued by stable expression of wildtype RNF170, whereas expression of RNF170 mutants (A.4 (c.396+3A>G, deletion of exon 5) and C.4 (c.304C>T, p.Cys102Arg)) fail to significantly restore IP3R-1 degradation. These mutation-specific analyses thus confirm our findings from the human patient fibroblasts (new Fig. 3).
- We did not observe significant accumulation of IP3R-1 in RNF170 deficient SH-SY5Y cells. These results contrast with our findings in fibroblasts and indicate differential regulatory mechanisms of basal IP3R levels in fibroblasts (IP3R-3) and neuronal cells (IP3R-1) (new Fig. 3).

Interestingly, the previously published dominant RNF170 mutation c.959C>T, leading to autosomal dominant sensory ataxia in humans, rescued IP3R-1 degradation in SH-SY5Y cells to a similar degree as wildtype RNF170, confirming a different mode of action for this variant. These results have also been added (new Fig. 3).

- Legends and figure layout sometimes poor – particularly the complex Fig 1.

For example:

- Links between legend and panels are not intuitive. For a start it would be easier to find the data referred to in each legend, if panels within each part of the figure, A, B, C, D, were explicitly subdivided as i, ii, iii, etc. Also many features in the legend are not explicitly labelled; many features in the figure are not explained in the legend, e.g. sizes of bands.
- Also, different parts of the figure are not clearly separated, e.g. B iii (in the above numbering) might easily belong to A iii.
- Fig. 1 and elsewhere: Use of +/+ for homozygous mutant individuals is counterintuitive and confusing for many readers: -/- is conventional in every other organism. If the authors wish to use this counterintuitive convention, they should explain it explicitly in their Fig legend.
- When coding regions are shown with sequencing chromatograms, please add DNA and amino acid coordinates to help the reader relate their data to their interpretation in the text.

Therefore: Please review the layout and legends of all figures and supplementary figures to improve the annotation and legends.

Reply: We are grateful for the reviewer's suggestions to make this complex figure more intuitive. In the revised version of the figure, the families are now clearly separated, the panels within each figure part are subdivided (i, ii, iii, ...), the description of the mutation

status was changed from +/- to wt/mut, and cDNA and amino acid coordinates were added to the electropherograms where possible (limitation: no representation of intronic variants in cDNA, e.g. Fig1 a.iii). We also changed the amino acid 1-letter-code to the 3-letter-code to make it easier to relate the variant descriptions in the text to the figure.

Fig 2d: Please add some indicator of variability to graph, e.g. SEM and N (and ideally individual datapoints if not too many)

Reply: The figure was updated according to the reviewer's suggestions. The revised figure (now Fig 2b) contains error bars indicating the standard deviation for each data point. The number of repetitions was added to the figure legend. In analogy to these changes, we also added the number of experiments to the other figures where applicable.

Fig 3 and Supp Fig 5: MO knockdown efficiency should be quantified. Also please state in Fig 3 legend what the control MO was.

Reply: We have now added details of the control morpholino in the materials and methods section and quantified MO knockdown efficiency in supplementary Fig 6c. Please note that the original Fig. 3 corresponds to Fig. 4 in the revised manuscript.

Fig 4 graphs. Please show N. And please replace bar graphs by graphs that show individual datapoints as in Fig 2b; mean and SEM (if normally distributed) can be superimposed on these as horizontal lines.

Reply: We have replaced the bars in Fig4D (now Fig. 5d) with data points and included N values for each group in the figure legend.

Abstract:

Line 68: those -> some

Reply: "those" was replaced by "some" as suggested.

Line 73

Introduction:

Line 84: Regulated Ca²⁺ release from the ER is mediated...

Reply: "Regulated" was added to the sentence as suggested.

Line 87: G-protein-coupled is hyphenated

Reply: The spelling was changed from "G protein coupled" to "G-protein-coupled" as suggested.

Reviewer #2 (Remarks to the Author):

This paper reports that loss of function mutations in RNF170 are linked with Hereditary Spastic Paraplegia (HSP) in several families. RNF170 is a ubiquitin E3 ligase, and InsP3R3 is one of its known targets. This genetic finding in itself is interesting and worth reporting, and genetic portion of the paper appears to be well done and convincing.

To explain the linkage between RNF170 and HSP, the authors argue that loss of function of RNF170 resulted in enhanced expression of InsP3Rs and abnormal calcium signaling in

cerebellar Purkinje cells and in other types of neurons as well. To support these claims, they include data with fibroblasts obtained from one of the families and demonstrate that stimulus-induced degradation of InsP3R3 in these cells is impaired and steady-state levels of InsP3R3 are elevated several fold (Fig 2). These are interesting data, but it is InsP3R1 and not InsP3R3 is a predominant neuronal isoform highly enriched in Purkinje cells. InsP3R1 in Purkinje cells does not undergo stimulus-dependent degradation, and in my opinion extrapolating the data with InsP3R3 in fibroblasts to InsP3R1 in Purkinje cells is too speculative. The authors have to establish how loss of RNF170 affects expression levels and function of cerebellar InsP3R1. Most likely such experiments will require creation and analysis of RNF170 knockout mice.

Reply: Patient fibroblasts offer the opportunity to study mutation effects in the relevant natural genetic background. However, we agree with the reviewer on the importance to transfer findings obtained in patient cells and thus the receptor isoform IP3R-3 to IP3R-1 and thus a neuronal model. We therefore used CRISPR/Cas9 to generate a homozygous knockout of RNF170 in the human neuronal cell line SH-SY5Y. Additionally, we performed rescue experiments with different RNF170 mutations (see above) in these cells. In these experiments we were able to demonstrate that RNF170 deficiency leads to impaired IP3R-1 degradation also in SH-SY5Y cells and that mutations observed in HSP patients fail to rescue this defect (see above and new Fig. 3). We would like to stress though that cerebellar Purkinje cells very likely are NOT the primarily affected cell type in RNF170 deficient patients; predominant and consistent spastic paraparesis in all patients indicate that RNF170 loss of function mutations instead primarily target upper motor neurons, thus leading to the phenotype of HSP.

They also present some data with Zebrafish, in which splicing of RNF170 was abrogated by morpholinos. Some interesting results were obtained, but these appear to be mostly developmental defects, the relevance of these findings for HSP is not clear. It is also not known if these defects are due to enhanced function of InsP3R or due to some other targets of RNF170. Genetic interaction experiments were not performed. It would have been informative to find out if InsP3R knockdown rescues RNF170 knockdown phenotype in Zebrafish. It was not done.

Reply: These are interesting questions and include *in vivo* experiments that we considered. However, we feel we have already substantially answered this with our *in vitro* RNF170 knockout and rescue experiments, proving that IP3R-1 degradation is regulated by RNF170 function. Our *in vivo* zebrafish overexpression data suggests that this pathway is under tight control, and additional early knockdown of *itpr1b* (the zebrafish IP3R1 orthologue) is only ever going to exacerbate the regulation of this pathway leading to confusing developmental defects. Unfortunately, at this time there is not a zebrafish specific ITPR1 antibody and despite trying some human antibodies we were unable to visualize any specific signal. The zebrafish data clearly shows that knockdown of *rnf170* causes neurodevelopmental defects, however we have now additionally included a later timepoint of study that shows progressive loss of neurons in muscle and brain. This is directly relevant to HSP, whereby it recapitulates the neuronal features associated with the disease.

They also perform some mutant RNF170 overexpression experiments in zebrafish embryos, but these are very difficult to interpret. If they wanted to test different RNF170 mutants perhaps better approach will be to perform rescue experiments with RNF170 KO SHY cells.

Reply: As stated above we have followed the excellent suggestion and performed rescue experiments in RNF170 knockout SH-SY5Y cells.

Overall, my opinion is that this paper potentially very interesting but not sufficiently developed. Genetic linkage with RNF170 LOF mutations is convincing. Proposed mechanism that involved upregulation of neuronal InsP3R1 function needs to be established further, most likely using RNF170 KO mouse model.

Reviewer #3 (Remarks to the Author):

This paper claims that inositol 1,4,5-trisphosphate signalling should be regarded as a key pathway for therapeutic intervention in hereditary spastic paraplegias and ataxias. Since ITPR1, ERLIN1 and ERLIN2 are already associated with these phenotypes, RNF170 expands the relevance of this important pathway to this disease category.

The conclusions of the paper are original in that they describe a new mode of inheritance and a slightly different phenotype than the one already associated with mutations in RNF170. This paper is of importance in the field since (1) it provides an independent confirmation of the involvement of RNF170 in neurodegenerative diseases (2) it associates mutations in RNF170 with an other mode of inheritance (3) it describes RNF170 mutations in a wider geographic and ethnic context (4) It expands on the biological validation of RNF170 and the understanding of its role in neurodegenerative diseases.

Reply: We respectfully disagree with Dr. Dupré that the phenotype we observe in our families is only 'slightly different' from the phenotype he and colleagues published in two Eastern Canadian families carrying the founder mutation c.595C>T. In contrast, we are convinced that the recessive loss-of-function mutations in RNF170 we report here have a different mode of action from the dominant c.959C>T and that this difference on the molecular level translates into a distinct selective vulnerability of neuronal systems and finally into a different phenotype. Pieces of evidence supporting this hypothesis are:

- The phenotype in RNF170-deficient HSP patients is characterized by infancy onset lower-limb predominant spastic tetraparesis, optic atrophy, cerebellar involvement (inconsistent) and peripheral neuropathy (inconsistent), thus indicating primary affection of upper motor neurons with inconsistent affection of the peripheral nervous system and cerebellar Purkinje cells. Central sensory tracts are only inconsistently affected at late disease stages. In contrast, the two families carrying the dominant c.595C>T variant in RNF170 demonstrate predominant affection of central sensory tracts leading to sensory ataxia with only subtle and rare affection of upper motor neurons (pyramidal signs in 3/17 reported cases, stretch reflexes reduced or absent, no case with manifest spasticity), no peripheral neuropathy and no cerebellar involvement (other than 'jerky saccades' described in 2/17 cases). Onset is adulthood/late adulthood and mode of inheritance autosomal dominant. Although the two diseases may look similar to a non-neurologist, as they both lead to gait and balance problems, the distinct involvement of neuronal systems clearly points towards distinct disease mechanisms.
- The molecular mechanism of action appears to be different between the dominant c.595C>T variant leading to ADSA and the recessive loss-of-function variants we describe here. While HSP-associated RNF170 variants lead to impairment of IP3R degradation (confirmed in patient fibroblasts and neuronal cells), no impact on RNF170 E3 ligase activity and IP3R ubiquitination was reported for the c.595C>T variant studied by Wright et al. 2015. In confirmation of these findings, expression of the c.595C>T variant in RNF170 knockout SH-SY5Y cells was able to rescue IP3R-1 degradation to a similar degree than expression of wildtype RNF170 (new Fig. 3).

We have rephrased the discussion to make the distinction between these two mutation types / phenotypes clearer.

Overall, RNF170 had been associated so far with a limited phenotype (autosomal dominant sensory ataxia, a disease of the cerebellum and the posterior columns). Following the publication of the present paper, it will become necessary to include RNF170 in HSP and ataxia panels, a key finding to maintain comprehensive clinical genetic testing.

The overall quality of the scientific work involved in this paper is very high, considering the standards required for the validation of genetic findings (WES, Sanger confirmation, segregation of variants, replication in other populations, fibroblasts, zebrafish model).

One critique is a matter of context and appraisal. References 31-32-33 are not mentioned in the introduction, and ADSA is not mentioned in the abstract. This omission does not allow the reader to place the findings in their context. RNF170 has been clearly shown to cause ataxia in the context of a spinal cord disease, and RNF170 is already known therefore as an important pathway for development of therapeutic interventions in this disease category.

Reply: We apologize for this omission. We have now added references 31-33 and mention of ADSA, previously only mentioned and discussed in detail in the discussion section to the introduction.

Lastly, Figure 5 would be better suited in a review paper.

Reply: Although we agree that Fig. 5 (now Fig. 6 in the revised manuscript) has some aspects of a 'review' to it, we still think it is valuable to give an immediate overview about the multitude of phenotypes associated with mutations in this pathway. As we think that it is important to highlight the wider implications and importance of IP3 signalling for cerebellar ataxias and motor neuron disease, we would actually suggest keeping the figure. Explaining this message in the text would certainly require much more space and be less clear.

Nicolas Dupré, MD, MSc, FRCP(C)
Neurologist & Clinician Scientist
CHU de Québec - UL
Associate Professor
Faculty of Medicine, Université Laval

Reviewer #4 (Remarks to the Author):

The manuscript "IP3 Receptor Degradation – A Mutational Hotspot for Hereditary Ataxia and Motor Neuron Disease" by M. Wagner is a well conducted genetic study with a initial functional characterization of the consequences of the identified mutations in cells and zebrafish.

The study is well presented and discussed and provides strong genetic evidence for the involvement of IP3 signaling in the pathophysiology of HSP. However there are one major and several some minor things to consider and some controls to include to validate and support the conclusions:

First and most importantly, the lack of a phenotype upon mRNA of the mutations t304c and truncated RNA could just simply reflect unstable RNA expression in zebrafish. Equal amount of protein translation needs to be shown here, eg by injection of tagged mRNAs and subsequent Western blot against the tag with quantification of the amount of protein translated.

Reply: The reviewer is correct. The lack of phenotype shown in the mutant RNA experiments could be a result of unstable RNA expression which is a likely consequence of the functional impact of the mutations.

Second, it should be pointed out that only about 50 of the transcripts are affected by the morpholinos.

Reply: Morpholinos can only ever be considered as a knockdown approach and the severity of the phenotype with only partial knockdown already shows compelling support for a functional role of the patient mutations in HSP.

Third, in figure 4 d the motor neurons seem to be oriented from posterior to anterior or the picture is aligned in a different way than the top 2 panels. The finding of more punctate staining with the ac tub antibody in the morpholino injected embryos could also reflect differences during fixation and should not be overstated.

Reply: In fact these figures are orientated in the same direction. This is now in supplementary fig 7, please observe the motile cilia in the pronephric ducts which form bright ciliary bundles in the proximal ducts (compared to observed single cilia in the distal tubule), these are consistent in each panel. The punctate staining is unlikely to be due to differences in fixation between controls and morphants as these were treated exactly the same way, and the results were consistent across multiple experiments performed on different days.

Fourth, phenocopy by overexpression of IPR3R-3 in zebrafish would be a nice addition.

Reply: This is an interesting suggestion, which we did investigate. However, we were unable to *in vitro* transcribe IP3R due to its lengthy nucleotide sequence (8.5kb).

Reviewers' Comments:

Reviewer #1:

Remarks to the Author:

The new data on the new KO line in Fig 3 addresses my main concern about the previous version, and I have only minor comments to add to my previous review.

Fig. 1a (i) It would be helpful to show chromatogram and sequence of wt/wt and mut/mut genomic sequence, if mut/mut is available from patient DNA.

Fig 2c. There don't appear to be 5 controls in this panel, as stated in legend.

Fig 2d. Mislabeled as 2b in legend

Fig. 3a and accompanying legend and text. The convention of ko/wt, ko/dellEx5, etc, is confusing. I first read it as heterozygous genotypes. Instead, to avoid this confusion, please use normal text for over expressed constructs, not superscript.

Fig 4 legend, line 646 - arborization

Supp Fig 3b - the c.304T arrow appears to be at the wrong position. Please check all such arrows throughout.

Reviewer #2:

Remarks to the Author:

In revised version of the paper the authors followed my suggestion and added results with SH-SY5Y cells with RNF170 KO. I would like to give them credit for making this effort, but unfortunately results that they obtain do not support their main hypothesis.

The data are shown on Fig 3a and 3b.

As it is clear from Fig 3b, RNF170 knockout does not increase steady-state levels of InsP3R1. If anything, there is a trend towards reduction in IP3R1 expression levels. In contrast, basal levels of InsP3R3 were increased 3 fold in RNF170 KO fibroblasts (Fig 2c).

On panel 3a they claim that there is a difference in stimulation-induced degradation of InsP3R1, but these data are not convincing as shown. Raw Western blot data are not presented for these experiments - compare for example with Fig 2a that supports stimulation-induced degradation of InsP3R3 in fibroblasts. There are 6 conditions shown on this panel, and it is very difficult to understand what changes are significant. They claim that InsP3R1 levels are increased almost 2-fold following carbachol stimulation in RNF170 KO cells. This seems to be the major difference from all other group. What is a mechanism for this increase? It can not be due to lack of degradation. There is a "rescue" of IP3R degradation at 2 h time point when RNF170 is re-expressed, but at 4 h time point InP3R1 expression levels are actually higher than at 2 h time point in this group. Mutants of RNF170 do not express at any significant amounts (Fig 3c) but InP3R1 levels in cells transfected with these mutants still significantly lower than in KO cells at 2h and 4 time points. Why? Overall, it appears that there was a lot of variability in Western blotting data used to generate Fig 3a and raw data are not shown.

It is well established that IP3R3 are actively degraded by proteasome following activation, but it is not the case for neuronal InsP3R1. This is a major problem for the proposed hypothesis, and data shown on Fig 3 are not able to address this concern.

My opinion regarding this paper remains the same - interesting genetic findings of RNF170 mutations linked to HSP, but the mechanistic link with InsP3R1 remains weak and not supported

by the data.

I also suggested genetic interaction studies in zebrafish to test linkage with InsP3R but these were not performed. The data that they have suggest that RNF170 plays some role in neuronal development, but it may not be related to InsP3R at all.

Reviewer #3:

Remarks to the Author:

No further comment.

Nicolas Dupré MD MSc FRCP

Neurologue

CHU de Québec - Université Laval

Professeur Agrégé

Faculté de médecine, Université Laval

Reviewer #4:

Remarks to the Author:

The resubmitted manuscript significantly improved by the addition of the second patient cell line as well as the CRISPR induced SH-SY5Y cells. This data strengthens the mechanistic link between RNF170 and IP3R-1.

I also appreciate the addition of the zebrafish analysis at a later time point upon KO of RNF170. However, based on the data provided, the two morpholinos have slightly different phenotypes which indicates unspecific toxicity. Rescue experiments with the RNF170 mRNA would exclude possible unspecific side effects and should be included. The mutant forms of RNF170 could then also be used to show reduced rescuing ability further strengthening the overexpression findings, that still lack quality controls in my opinion.

Reviewers' comments:

Reviewer #1 (Remarks to the Author):

The new data on the new KO line in Fig 3 addresses my main concern about the previous version, and I have only minor comments to add to my previous review.

Fig. 1a (i) It would be helpful to show chromatogram and sequence of wt/wt and mut/mut genomic sequence, if mut/mut is available from patient DNA.

Reply: Done as suggested.

Fig 2c. There don't appear to be 5 controls in this panel, as stated in legend.

Reply: We apologize for the mistake and corrected Fig 2c and the legend accordingly.

Fig 2d. Mislabeled as 2b in legend

Reply: Corrected accordingly.

Fig. 3a and accompanying legend and text. The convention of ko/wt, ko/dellEx5, etc, is confusing. I first read it as heterozygous genotypes. Instead, to avoid this confusion, please use normal text for over expressed constructs, not superscript.

Reply: We have followed the suggestion of the reviewer and renamed the overexpression cell lines in text and figures, hereby avoiding the use of superscript. New nomenclature is SH-SY5Y(RNF170^{ko} (Δ Ex5-HA)) for re-expression of RNF170 Δ Ex5 in a knockout background and accordingly. Since the figure has also been revised in response to the comments of reviewer #2, the results that were in Fig. 3a are now in Fig. 3d and are limited to the knockout model, the wildtype and the rescue experiment.

Fig 4 legend, line 646 – arborization

Reply: Corrected accordingly.

Supp Fig 3b - the c.304T arrow appears to be at the wrong position. Please check all such arrows throughout.

Reply: We have checked all arrows and they are at the correct position now.

Reviewer #2 (Remarks to the Author):

In revised version of the paper the authors followed my suggestion and added results with SH-SY5Y cells with RNF170 KO. I would like to give them credit for making this effort, but unfortunately results that they obtain do not support their main hypothesis. The data are shown on Fig 3a and 3b.

As it is clear from Fig 3b, RNF170 knockout does not increase steady-state levels of InsP3R1. If anything, there is a trend towards reduction in IP3R1 expression levels. In contrast, basal levels of InsP3R3 were increased 3 fold in RNF170 KO fibroblasts (Fig 2c).

Reply: In the previous Fig 3b, there was a trend towards an **increased** basal IP3R level in SH-SY5Y cells (and not a reduction as the reviewer states). As we observed a trend towards increased basal IP3R levels in SH-SY5Y cells, we doubled the number of biological replicates to 8; differences between wt and ko cells are now significant, clearly demonstrating increased basal IR3R-1 levels in RNF170 deficient SH-SY5Y cells compared to wt (Fig 3b). For a more

logic composition, we have restructured the respective section (lines 236 – 249). This finding is in line with the results observed in fibroblasts.

On panel 3a they claim that there is a difference in stimulation-induced degradation of InsP3R1, but these data are not convincing as shown. Raw Western blot data are not presented for these experiments - compare for example with Fig 2a that supports stimulation-induced degradation of InsP3R3 in fibroblasts. There are 6 conditions shown on this panel, and it is very difficult to understand what changes are significant. They claim that InsP3R1 levels are increased almost 2-fold following carbachol stimulation in RNF170 KO cells. This seems to be the major difference from all other group. What is a mechanism for this increase? It cannot be due to lack of degradation.

Reply: We agree that Fig 3a was very difficult to understand with 6 conditions included. Following the suggestion of the reviewer, we added representative Western blot raw data to the figure (Fig 3a, c and e) (all original blots are included in the source data file provided with the submission).

Fig. 3d now only included 3 conditions: wt SH-SY5Y, RNF170 ko SH-SY5Y and re-expression of wt RNF170 in a knockout background ("wildtype-rescue"). P-values are clearly stated in the text. The results from re-expression of RNF170 mutants (Δ Ex5, 304T>C, 595C>T) were removed as these mutant proteins do not reach significant RNF170 expression levels as demonstrated in Fig. 3g.

We further added additional biological replicates to substantiate our findings. Our data now demonstrate:

- Basal levels of IP3R-1 are increased in RNF170 deficient SH-SY5Y cells (Fig. 3a+b), in line with our findings in patient fibroblasts and published literature (see manuscript text lines 119-127)
- The accumulation of IP3R-1 in RNF170 deficient SH-SY5Y cells can be rescued by re-expression of wildtype RNF170 (Fig. 3e+f). This supports a causal relationship between the loss of RNF170 function and the observed IP3R-1 accumulation in knockout cells.
- Carbachol-dependent IP3R-1 degradation is variable and partial even in SH-SY5Y cells expressing wildtype RNF170 (Fig. 3c+d), at least in our hands. Although we observe a trend towards loss of stimulus-dependent IP3R-1 degradation that can be rescued by re-expression of wildtype RNF170, these findings are not statistically significant. We therefore interpreted these findings with caution in the manuscript text.

Taken together, these findings suggest an accumulation of IP3R which might or might not be stimulation dependent as a consequence of RNF170 dysfunction. Findings in SH-SY5Y cells are in accordance with findings in patient fibroblasts. We therefore thank the reviewer for his thorough evaluation of our data and corrected our conclusions throughout the manuscript (lines 241 – 249, 316 – 317 and 321 – 328).

There is a "rescue" of IP3R degradation at 2 h time point when RNF170 is re-expressed, but at 4 h time point InP3R1 expression levels are actually higher than at 2 h time point in this group. Mutants of

RNF170 do not express at any significant amounts (Fig 3c) but InP3R1 levels in cells transfected with these mutants still significantly lower than in KO cells at 2h and 4 time points. Why? Overall, it appears that there was a lot of variability in Western blotting data used to generate Fig 3a and raw data are not shown.

Reply: As mentioned above, we have included WB data that was used to generate the figures shown. Western Blot per se is a semi-quantitative method and was not used to determine exact protein levels but to compare different conditions and observe differences between conditions. The reviewer is right when he claims that there is significant variability, which, however, reflects the actual findings. As we did not observe a significant difference in stimulation-induced degradation when adding further biological replicates to our experiments, we adjusted our conclusions which we draw from the data accordingly (lines 241 – 249, 321 – 328).

It is well established that IP3R3 are actively degraded by proteasome following activation, but it is not the case for neuronal InsP3R1. This is a major problem for the proposed hypothesis, and data shown on Fig 3 are not able to address this concern. My opinion regarding this paper remains the same - interesting genetic findings of RNF170 mutations linked to HSP, but the mechanistic link with InsP3R1 remains weak and not supported by the data.

Reply: A large body of literature confirms the stimulus dependent and ubiquitin-proteasome mediated degradation of all three IP3R subtypes including IP3R-1 in various cell types including neuronal cells (e.g. ^{1, 2, 3, 4, 5}, reviewed in ⁶). However, the specific role of RNF170 and the ERAD pathway in stimulation-induced degradation vs. basal turnover of IP3R still has to be elucidated. As this difference was previously not specifically addressed, we added background information to the introduction (lines 119 – 122). While we clearly demonstrate that loss of RNF170 increases basal levels of IP3R (therefore establishing a mechanistic link) both in fibroblasts and SH-SY5Y cells, we were not able to delineate whether this effect is due to impaired stimulus-dependent degradation of IP3R-1 in SH-SY5Y cells or due to impaired basal turnover or both. This limitation is now discussed in the manuscript (lines 321 – 328).

I also suggested genetic interaction studies in zebrafish to test linkage with InsP3R but these were not performed. The data that they have suggest that RNF170 plays some role in neuronal development, but it may not be related to InsP3R at all.

Reply: We agree that the zebrafish data on its own does not prove that RNF170 causes impaired neuronal development via IP3R degradation. Following the initial request for genetic interaction studies we suggested that as “Our in vivo zebrafish overexpression data suggests that this pathway is under tight control, additional early knockdown of *itpr1b* (the zebrafish IP3R1 orthologue) is only ever going to exacerbate the regulation of this pathway leading to confusing developmental defects”, (please see also our response to reviewer 4 below). In fact it is already known that decreased levels of ITPR1 result in a movement disorder as seen in

patients with loss of function mutations in ITPR1 and in the mouse mutants^{7, 8, 9, 10, 11, 12}, thereby making it very difficult to interpret the results of a double knockdown. However, as we have shown that loss of RNF170 leads to an accumulation of IP3R in fibroblasts and neuronal cells, a finding which is in line with previous reports (as cited in the manuscript) we believe that this is the most likely mechanism even though it cannot be excluded that other pathways play a significant role. We have adjusted the language in our manuscript to reflect this uncertainty.

Reviewer #4 (Remarks to the Author):

The resubmitted manuscript significantly improved by the addition of the second patient cell line as well as the CRISPR induced SH-SY5Y cells. This data strengthens the mechanistic link between RNF170 and IP3R-1.

I also appreciate the addition of the zebrafish analysis at a later time point upon KO of RNF170. However, based on the data provided, the two morpholinos have slightly different phenotypes which indicates unspecific toxicity. Rescue experiments with the RNF170 mRNA would exclude possible unspecific side effects and should be included. The mutant forms of RNF170 could then also be used to show reduced rescuing ability further strengthening the overexpression findings that still lack quality controls in my opinion.

Reply: The reviewer suggests that the two morpholinos (MOs) give slightly different phenotypes which would indicate unspecific toxicity, but we have to respectfully disagree, since the two morpholinos give a remarkably similar phenotype. At 48 hpf they are indistinguishable and show comparable general morphology and neurological defects (Fig 4). At 5 dpf, whilst one MO shows remnants of the motoneurons the second MO is more severe. These differences are unlikely to be due to MO toxicity since we control against this by using a non-specific morpholino (against an intronic region of human B-globin, absent in zebrafish) at the same working concentration as the gene targeted MOs. It is more likely that the differences are due to subtle changes in MO efficiency between the two, which we have shown in supplementary Fig 6.

We thank the reviewer for suggesting the rescue experiment to further prove specificity. We have attempted this experiment but find overexpression of RNF170 results in exacerbation of the developmental phenotype. As such we have been unable to titrate our rescue construct to a level that ameliorates the defects, without causing a strong overexpression phenotype. This is a problem commonly observed within the zebrafish research community for endogenous genes subject to exquisitely specific and complex spatial-temporal regulation¹³. Indeed, we demonstrate (Fig 4) that ectopic RNF170 expression leads to developmental abnormalities in control embryos, so it is not surprising that loss of endogenous *rnf170* and addition of ectopic RNF170 might lead to a more severe phenotype. For genes that are expressed in a restricted manner, or that have a strong overexpression phenotype, obtaining a true rescue may be difficult or impossible¹⁴. In addition, RNA diffusion into the developing cell is less efficient than

MOs, which means distribution of the two can differ, again making rescue experiments difficult to implement^{15, 16}.

To provide further evidence for MO specificity, we used an additional third nonoverlapping MO instead. Knockdown resulted in a markedly similar phenotype as shown in supplementary Fig 8 as well as in supplementary movies (d) and (e).

1. Wojcikiewicz RJ. Type I, II, and III inositol 1,4,5-trisphosphate receptors are unequally susceptible to down-regulation and are expressed in markedly different proportions in different cell types. *J Biol Chem* **270**, 11678-11683 (1995).
2. Oberdorf J, Webster JM, Zhu CC, Luo SG, Wojcikiewicz RJ. Down-regulation of types I, II and III inositol 1,4,5-trisphosphate receptors is mediated by the ubiquitin/proteasome pathway. *Biochem J* **339 (Pt 2)**, 453-461 (1999).
3. Zhu CC, Wojcikiewicz RJ. Ligand binding directly stimulates ubiquitination of the inositol 1, 4,5-trisphosphate receptor. *Biochem J* **348 Pt 3**, 551-556 (2000).
4. Sliter DA, Kubota K, Kirkpatrick DS, Alzayady KJ, Gygi SP, Wojcikiewicz RJ. Mass spectrometric analysis of type 1 inositol 1,4,5-trisphosphate receptor ubiquitination. *J Biol Chem* **283**, 35319-35328 (2008).
5. Wojcikiewicz RJ, Furuichi T, Nakade S, Mikoshiba K, Nahorski SR. Muscarinic receptor activation down-regulates the type I inositol 1,4,5-trisphosphate receptor by accelerating its degradation. *J Biol Chem* **269**, 7963-7969 (1994).
6. Wright FA, Wojcikiewicz RJ. Chapter 4 - Inositol 1,4,5-Trisphosphate Receptor Ubiquitination. *Prog Mol Biol Transl Sci* **141**, 141-159 (2016).
7. Marelli C, *et al.* SCA15 due to large ITPR1 deletions in a cohort of 333 white families with dominant ataxia. *Arch Neurol* **68**, 637-643 (2011).
8. Matsumoto M, *et al.* Ataxia and epileptic seizures in mice lacking type 1 inositol 1,4,5-trisphosphate receptor. *Nature* **379**, 168-171 (1996).
9. Street VA, *et al.* The type 1 inositol 1,4,5-trisphosphate receptor gene is altered in the opisthotonos mouse. *J Neurosci* **17**, 635-645 (1997).
10. van de Leemput J, *et al.* Deletion at ITPR1 underlies ataxia in mice and spinocerebellar ataxia 15 in humans. *PLoS Genet* **3**, e108 (2007).
11. Das J, Lilleker J, Shereef H, Ealing J. Missense mutation in the ITPR1 gene presenting with ataxic cerebral palsy: Description of an affected family and literature review. *Neurol Neurochir Pol* **51**, 497-500 (2017).
12. Huang L, *et al.* Missense mutations in ITPR1 cause autosomal dominant congenital nonprogressive spinocerebellar ataxia. *Orphanet J Rare Dis* **7**, 67 (2012).
13. Piepenburg O, Grimmer D, Williams PH, Smith JC. Activin redux: specification of mesodermal pattern in *Xenopus* by graded concentrations of endogenous activin B. *Development* **131**, 4977-4986 (2004).
14. Eisen JS, Smith JC. Controlling morpholino experiments: don't stop making antisense. *Development* **135**, 1735-1743 (2008).
15. Nutt SL, Bronchain OJ, Hartley KO, Amaya E. Comparison of morpholino based translational inhibition during the development of *Xenopus laevis* and *Xenopus tropicalis*. *Genesis* **30**, 110-113 (2001).
16. Saka Y, Smith JC. A *Xenopus* tribbles orthologue is required for the progression of mitosis and for development of the nervous system. *Dev Biol* **273**, 210-225 (2004).

Reviewers' Comments:

Reviewer #2:

Remarks to the Author:

The authors made serious effort to clean up SH-SY5Y cells data (Fig 3). In the present form these data are more clear. Effects on InsP3R1 levels are mild, but it does appear to be an increase in InsP3R1 levels following RNF170 KO (Fig 3). It is a relatively mild increase in InsP3R1 expression levels (about 50%) and most likely it should not result in any significant changes in cell physiology. But at least it is consistent with the proposed idea.

Data with carbachol stimulation (Fig 3d) are more confusing. There is some degradation of InsP3R1 in WT cells at 2 h point, but it "recovers" (how?) at 4 h time point. In RNF170KO cell levels of InsP3R1 increase (?) following charbacol stimulation. This is NOT degradation defect as stated in the paper, it is actual INCREASE in InsP3R1 levels that authors do not have explanation for.

Panels 3e and 3f are basically repeat of panels 3a and 3b with addition of rescue experiment. These data are redundant.

Here is my overall impression. The paper reports that mutations in RNF170 are linked to Hereditary Spastic Paraplegia. This is a novel and interesting finding. They further propose that these mutations are loss of function mutations that impair activation-dependent degradation of InsP3Rs. They have data supporting this claim in experiments with patient fibroblasts that express InsP3R3 (Fig 2). They made serious effort trying to demonstrate the same phenomenon in SH-SY5Y cells that express InsP3R1 (neuronal isoform), but these data are much less impressive. My suggestion is to leave just panels Fig 3e, 3f and 3g. Remove panels a and b because they are redundant and panels c and d because these data are confusing and non-convincing.

They toned down their language in the revision, and I think the paper can be published after Fig 3 is changed as suggested above. I am still not convinced that RNF170 mutations act by impairing activity-dependent degradation of InsP3R1 in neurons. But as discussed above they have a few lines of supporting evidence and readers can decide themselves if they are convinced by these data (I am not)

Reviewer #4:

Remarks to the Author:

I still believe the paper provides great genetic data pointing to RNF170 and IP3R as a cause for HSP and are worth being published. However, I still have some concerns which could be simply addressed or rephrased in the current manuscript version.

1. Morpholino experiments are still very critically evaluated in the zebrafish community and need to be performed with appropriate controls to draw conclusions. Crispr/Cas9 Mutants are the current standard and can be very powerful in combination with KDs. Failure to rescue is (as stated correctly in the rebuttal) sometimes observed due to high toxicity of the injected mRNA, but in this case leaves the experiments of the KD without proper controls. Adding new morpholino does not compensate for the required quality standards. This needs to at least be discussed in the manuscript.
2. It has been nicely shown that mutant RNA is less stable in cells and patient derived cell cultures. There are absolutely now quality controls for the overexpression experiments in zebrafish. Is the RNA also less stable there? What are the amounts of RNA expressed in the mutants (WB of tagged mRNAs of injected embryos). Without quantification of input into the embryo and resulting protein expression, these experiments are difficult to interpret and should be

discussed with more caution.

3. In line 280 “displayed reduced and absent MNs suggesting progressive MN degeneration” is claimed. What I see in Suppl. Fig. 7c is absolutely no staining of acetulated tubulin upon E3MO, more suggestive of failed staining than loss of MN! There is no staining visible in the spinal cord as for E4MO as well on this picture which would have been a nice internal control for the staining. The MN stainings are so drastically different in both morpholino experiments even though the KD efficiency seems comparable on RNA level. Please check orientation of lowest panel I Suppl. Fig. 7b

4. While the authors are showing developmental defects of the zebrafish larvae upon KD, I find the statement in line 288 overstated: “recapitulates many of the clinical features observed in the HSP patients”. At no point is the embryo shown to be normal and I would rather describe the effects developmental rather than degenerative.

5. In line 126 of Suppl. Fig 8 it is stated that “impaired motorneuron formation”. To me it looks like they are formed but stain differently at 48hpf. Please clarify and state correctly.

Reviewer #4 (Remarks to the Author):

I still believe the paper provides great genetic data pointing to RNF170 and IP3R as a cause for HSP and are worth being published. However, I still have some concerns which could be simply addressed or rephrased in the current manuscript version.

1. Morpholino experiments are still very critically evaluated in the zebrafish community and need to be performed with appropriate controls to draw conclusions. Crispr/Cas9 Mutants are the current standard and can be very powerful in combination with KDs. Failure to rescue is (as stated correctly in the rebuttal) sometimes observed due to high toxicity of the injected mRNA, but in this case leaves the experiments of the KD without proper controls. Adding new morpholino does not compensate for the required quality standards. This needs to at least be discussed in the manuscript.

Reply: We agree with reviewer #4 and have discussed that rescue experiments failed which is why we cannot prove specificity of the morpholino knockdown (lines 291-293). We believe that these clear statements leave the reader to interpret the knockdown findings unbiased.

2. It has been nicely shown that mutant RNA is less stable in cells and patient derived cell cultures. There are absolutely no quality controls for the overexpression experiments in zebrafish. Is the RNA also less stable there? What are the amounts of RNA expressed in the mutants (WB of tagged mRNAs of injected embryos). Without quantification of input into the embryo and resulting protein expression, these experiments are difficult to interpret and should be discussed with more caution.

Reply: We have toned down our language and discussed that we cannot exclude the possibility that these differences are due to reduced RNA stability rather than aberrant protein function of the mutant RNF170 (lines 354-356).

3. In line 280 “displayed reduced and absent MNs suggesting progressive MN degeneration” is claimed. What I see in Suppl. Fig. 7c is absolutely no staining of acetylated tubulin upon E3MO, more suggestive of failed staining than loss of MN! There is no staining visible in the spinal cord as for E4MO as well on this picture which would have been a nice internal control for the staining. The MN stainings are so drastically different in both morpholino experiments even though the KD efficiency seems comparable on RNA level. Please check orientation of lowest panel I Suppl. Fig. 7b

Reply: In our initial Supplementary Figure 7 we had included the most severe phenotype observed. We agree that this image could be due to failed staining and have replaced it by a more suitable image. In concordance, we have changed „displayed reduced and absent MNs“ to „Morphants displayed persistent reduction in MN staining“ in the manuscript.

4. While the authors are showing developmental defects of the zebrafish larvae upon KD, I find the statement in line 288 overstated: “recapitulates many of the clinical features observed in the HSP patients”. At no point is the embryo shown to be normal and I would rather describe the effects developmental rather than degenerative.

Reply: We have changed our statement that the zebrafish model recapitulates many of the clinical HSP features and toned it down as suggested by the reviewer (line 298-299). We have also omitted the statement that the effects of RNF170 KD are neurodegenerative (line 289).

5. In line 126 of Suppl. Fig 8 it is stated that “impaired motoneuron formation”. To me it looks like they are formed but stain differently at 48hpf. Please clarify and state correctly.

Reply: We have removed our statement claiming impaired motoneuron formation.